# SPEN is required for *Xist* upregulation during initiation of X chromosome inactivation

Teresa Robert-Finestra [1], Beatrice F. Tan [1,6], Hegias Mira-Bontenbal[1,6], Erika Timmers[1,6], Cristina Gontan[1], Sarra Merzouk[1], Benedetto Daniele Giaimo[2], François Dossin [3], Wilfred F. J. van IJcken [4], John W. M. Martens [5], Tilman Borggrefe[2], Edith Heard [3] & Joost Gribnau [1✉]

At initiation of X chromosome inactivation (XCI), *Xist* is monoallelically upregulated from the future inactive X (Xi) chromosome, overcoming repression by its antisense transcript *Tsix*. *Xist* recruits various chromatin remodelers, amongst them SPEN, which are involved in silencing of X-linked genes in *cis* and establishment of the Xi. Here, we show that SPEN plays an important role in initiation of XCI. *Spen* null female mouse embryonic stem cells (ESCs) are defective in *Xist* upregulation upon differentiation. We find that *Xist*-mediated SPEN recruitment to the Xi chromosome happens very early in XCI, and that SPEN-mediated silencing of the *Tsix* promoter is required for *Xist* upregulation. Accordingly, failed *Xist* upregulation in *Spen*$^{-/-}$ ESCs can be rescued by concomitant removal of *Tsix*. These findings indicate that SPEN is not only required for the establishment of the Xi, but is also crucial in initiation of the XCI process.

[1] Department of Developmental Biology, Erasmus University Medical Center, Oncode Institute, 3015GD Rotterdam, The Netherlands. [2] Institute of Biochemistry, University of Giessen, 35392 Giessen, Germany. [3] European Molecular Biology Laboratory, Director's Research, 69117 Heidelberg, Germany. [4] Center for Biomics, Erasmus University Medical Center, 3015CN Rotterdam, The Netherlands. [5] Department of Medical Oncology, Erasmus MC Cancer Institute and Cancer Genomics Netherlands, Erasmus University Medical Center, 3015CN Rotterdam, The Netherlands. [6] These authors contributed equally: Beatrice F. Tan, Hegias Mira-Bontenbal, Erika Timmers. ✉email: j.gribnau@erasmusmc.nl

To compensate for gene-dosage imbalance between females (XX) and males (XY), female placental mammals randomly inactivate one X chromosome early during embryonic development[1]. In mice, random X-chromosome inactivation (XCI) takes place in the epiblast in three phases: initiation, establishment, and maintenance. During the initiation phase, the long noncoding RNA (lncRNA) Xist is upregulated from the future inactive X (Xi) chromosome[2–4]. Xist is located within the X Chromosome Inactivation Centre (Xic), an X-linked region required for XCI that contains different cis-regulatory elements, including Tsix, another lncRNA gene that is transcribed antisense to and completely overlaps Xist[5]. Tsix negatively regulates Xist expression via antisense transcription and chromatin remodeling[6–12]. Together, Xist and Tsix form a master switch that controls initiation of XCI: in the pluripotent state, Tsix is biallelically expressed, repressing both alleles of Xist, while upon initiation of XCI, the balance changes, resulting in downregulation of Tsix and upregulation of Xist on the future Xi. Trans-regulators including pluripotency factors OCT4, NANOG, REX1 as well as XCI activators are key in controlling this switch by regulating Xist and Tsix transcription (reviewed in[13]). Later, during the establishment phase of XCI, the 17 kb lncRNA Xist spreads in cis along the future Xi and recruits different proteins involved in gene silencing that render the X transcriptionally inactive (reviewed in[14]). Active histone marks (H3K4me2/me3, H3 and H4 acetylation) are removed and repressive histone marks are instigated, catalyzed by polycomb group complexes and other protein complexes (reviewed in[15]). Finally, in the maintenance phase, the inactive state of the Xi is epigenetically propagated across cell divisions.

Different studies consisting of Xist RNA immunoprecipitations coupled to mass spectrometry[16–18] and genetic screens[19,20] identified SPEN (also known as SHARP in human and MINT in mouse) as a crucial factor in the establishment phase of XCI. SPEN is a large protein with four N-terminal RNA recognition motifs (RRM) and a highly conserved C-terminal SPOC domain able to recruit different proteins involved in transcriptional silencing[21,22]. SPEN is also involved in the Notch signaling pathway and nuclear receptor signaling, where it acts as a transcriptional corepressor[23,24].

SPEN is crucial for X-linked gene silencing[16,17,19,20] by binding the Xist repeat A (RepA) via its RRM domains[17,19] and interacting via its SPOC domain with the corepressors NCoR/SMRT to recruit/activate histone deacetylase 3 (HDAC3), responsible for the removal of histone H3 and H4 acetylation at promoters and enhancers of genes located on the future Xi[16,25]. Despite this crucial role for SPEN in establishment of the Xi, these studies did not report defects in Xist upregulation and coating[16,17,19,20,25], possibly due to forced Xist upregulation using doxycycline-inducible systems (Supplementary Table 1).

Here, we show that SPEN accumulates on the Xi very early during differentiation and is required for Xist upregulation. We show that SPEN has a dual function, required to silence Tsix and facilitate Xist upregulation, and stabilization of Xist RNA. Together, our results indicate that SPEN is not only necessary for X-linked gene silencing, but also plays a crucial earlier role in the regulation of initiation of XCI.

## Results

**SPEN is required for Xist upregulation.** Previous work has shown how SPEN is crucial for silencing of X-linked genes, but these studies did not investigate the role of SPEN in initiation of Xist expression. Therefore, we generated Spen homozygous (Spen[−/−]) and heterozygous (Spen[+/−]) knockout mouse embryonic stem cells (ESCs) by deleting the complete open-reading frame (ORF) using

the CRISPR/Cas9 technology (Supplementary Fig. 1a). These lines were generated in a hybrid F1 129/Sv:Cast/EiJ (129/Cast) genetic background with a doxycycline-responsive endogenous Xist promoter located on the Cast X chromosome[26] (Fig. 1a). The Spen ORF deletion was verified by PCR on genomic DNA (gDNA) and Western blot analysis (Supplementary Fig. 1b,c). Allele-specific RNA-seq analysis (Supplementary Fig. 1d) of wild-type (Wt) undifferentiated ESCs (day 0) containing the doxycycline-responsive Xist promoter treated with and without doxycycline for 4 days showed skewed X-linked gene silencing toward the Cast allele (Fig. 1b, Supplementary Fig. 1e top left). On the other hand, the same analysis in Spen[−/−] ESCs showed impaired X-linked gene silencing (Fig. 1b, Supplementary Fig. 1e bottom left), as described before[16,17,19,20] (Supplementary Table 1).

To trigger XCI in the context of differentiation, we forced Xist upregulation by doxycycline treatment followed by monolayer differentiation. Allele-specific RNA-seq analysis of Wt and Spen[−/−] ESCs treated with doxycycline at day 7 of monolayer differentiation (Supplementary Fig. 1d) also revealed a lack of X-linked gene silencing along the entire X chromosome (Fig. 1b, Supplementary Fig. 1e bottom right). Similarly, allele-specific RT-qPCR analysis of the X-linked gene Rnf12 revealed impaired silencing (Fig. 1c). Although previous work indicated that in Spen[−/−] ESCs, a group of lowly expressed X-linked genes are susceptible to SPEN-independent gene silencing[27], our RNA-seq analysis shows no silencing of this specific group of genes (Supplementary Fig. 1f,g). We observed Xist upregulation in Wt and Spen[−/−] cells, however doxycycline induction resulted in lower Xist expression levels in Spen[−/−] compared with Wt ESCs (Fig. 1d), an effect that was also reported in a recent study[27].

Next, we recapitulated physiological XCI by monolayer differentiation in the absence of doxycycline to allow normal Xist upregulation and X-linked gene silencing. Allele-specific RNA-seq analysis of Wt and Spen[−/−] ESCs at day 3 without doxycycline (Supplementary Fig. 1d) shows a significant difference in silencing (Fig. 1b, Supplementary Fig. 1e top right) also visible for Rnf12 at day 3 by RT-qPCR and more pronounced at later time points of differentiation (days 5 and 7) (Fig. 1e). Remarkably, in Spen[−/−] cells, Xist upregulation from the 129 allele was completely abrogated (Fig. 1f), contrasting earlier evidence that suggested that SPEN is not required for Xist upregulation and coating[16,17,19,20,25] (Supplementary Table 1). In addition, differentiating Spen[−/−] ESCs lack Xist clouds, determined by RNA-FISH (Fig. 1g,h), while Tsix was significantly more expressed from the Wt 129 allele in Spen[−/−] cells compared with Wt cells in differentiating cells (Fig. 1i), suggesting that SPEN might be necessary for Tsix silencing. Importantly, RT-qPCR analysis confirmed proper silencing of pluripotency genes Rex1 and Nanog, and upregulation of endoderm marker Gata6 (Supplementary Fig. 1h) in Spen[−/−] ESCs upon monolayer differentiation, indicating that loss of Xist expression is not related to defective ESC differentiation.

Furthermore, while Spen[+/−] cells are able to upregulate Xist and silence Rnf12 upon doxycycline treatment followed by monolayer differentiation (Supplementary Fig. 2a,b), they show reduced Rnf12 silencing upon physiological differentiation without a defect in Xist upregulation (Supplementary Fig. 2c,d). Given that SPEN levels are reduced in Spen[+/−] cells (Supplementary Fig. 1c), these results demonstrate that SPEN dosage is important in XCI.

**Spen rescue leads to normal Xist expression levels.** To confirm our results, we performed a rescue experiment by stably reexpressing Spen through introduction of the full-length Spen cDNA in the ROSA26 locus[25] of Spen[−/−] ESCs (Supplementary Fig. 3a).

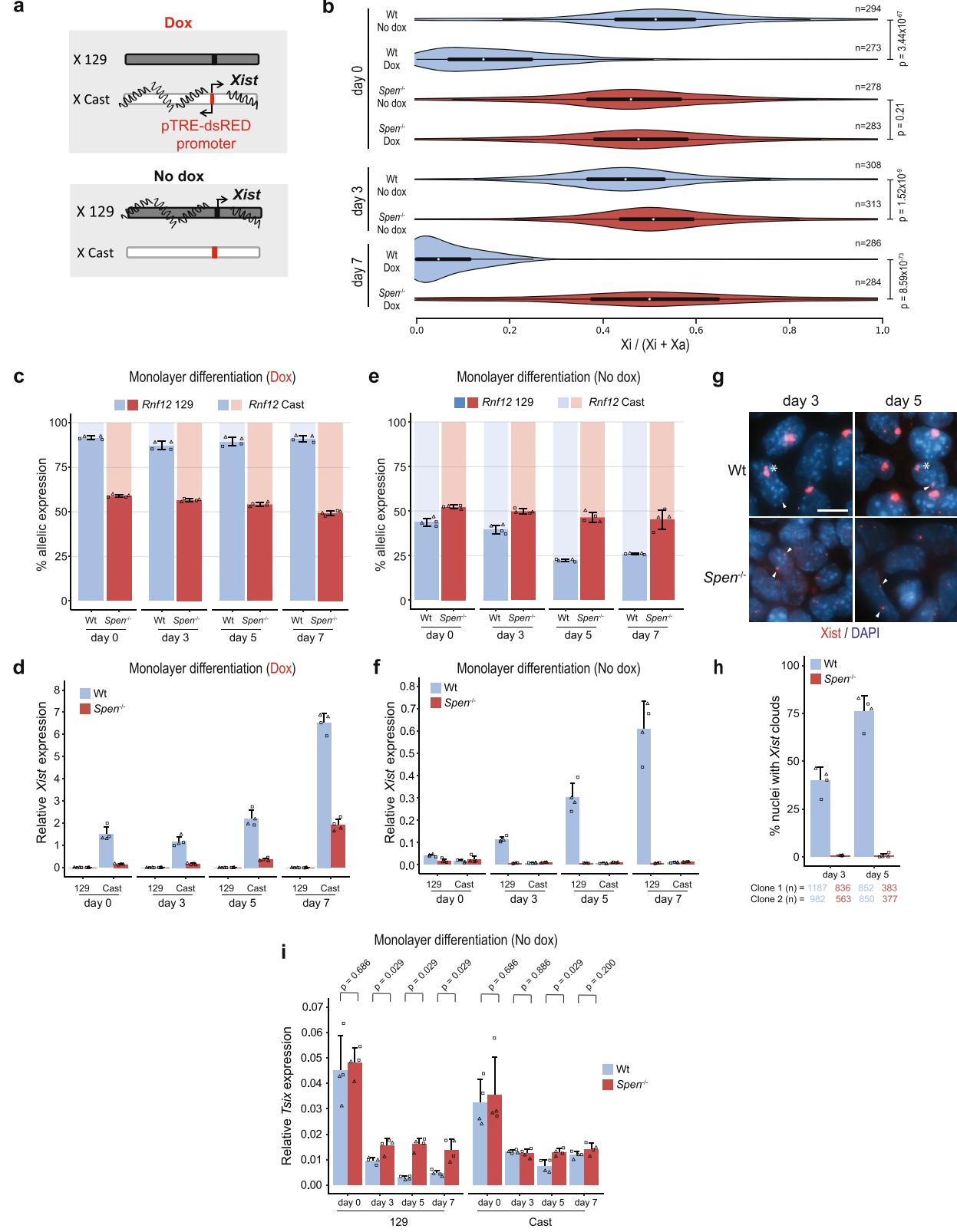

Successful integration was verified by gDNA PCR and confirmed by *Spen* mRNA and protein levels by RT-qPCR and Western blot, respectively (Supplementary Fig. 3b–d). The generated rescue clones (Clones A, B, and C) display two- to three-fold over-expression of *Spen* RNA at the ESC stage and during monolayer differentiation (Supplementary Fig. 3c). In contrast to the *Spen*$^{-/-}$ lines, all *Spen*$^{-/-(cDNA)}$ rescue clones express *Xist* at similar levels to Wt clones during monolayer differentiation (Fig. 2a). Allele-specific expression analysis of *Rnf12* and *Tsix* (129 allele) indicated that the silencing defect is partially rescued in *Spen*$^{-/-(cDNA)}$ clones (Supplementary Fig. 3e, f). Moreover, RT-qPCR analysis shows silencing of pluripotency markers (*Sox2* and *Oct4*) and upregulation of the endoderm marker *Gata6* in the *Spen* rescue clones (Supplementary Fig. 3g), indicating that *Spen* overexpression

**Fig. 1 Impaired *Xist* upregulation in *Spen*[−/−] ESCs upon monolayer differentiation. a** Overview of the endogenous doxycycline-inducible *Xist* hybrid system used in this study. Addition of doxycycline leads to inactivation of the Cast X chromosome, while no addition of the drug leads to inactivation of the 129 X chromosome. **b** Violin plots depicting the distribution of the allelic ratios ((Xi)/(Xi+Xa)) of X-linked genes in Wt and *Spen*[−/−] ESCs, untreated or treated with doxycycline at days 0, 3, and 7 of differentiation. This figure summarizes data in Supplementary Fig. 1e. The box plots inside the violin plots display the median (central white dot), the interquartile range (box limits) and the whiskers represent 1.5x of the interquartile range. Two-sided Mann–Whitney test corrected with Benjamini–Hochberg for multiple testing (α < 0.05). *n* = genes with more than 20 reads. Xa = active X chromosome. **c** Percentage of *Rnf12* allelic expression at different time points of monolayer differentiation of two independent Wt and *Spen*[−/−] ESC lines treated with doxycycline, determined by RT-qPCR. Relative *Rnf12* allelic (129 and Cast) expression was normalized to *Rnf12* total expression and averaged ± standard deviation (SD). Reference gene: *Hist2h2aa1*. **d** Relative allele-specific *Xist* expression of two independent Wt and *Spen*[−/−] ESC lines at different time points of monolayer differentiation treated with doxycycline. Average expression ± SD. Reference gene: *Hist2h2aa1*. **e** Same as displayed in (**c**) without doxycycline. **f** Same as displayed in (**d**) without doxycycline. **g** *Xist* RNA FISH (red) of Wt and *Spen*[−/−] ESC lines at days 3 and 5 of differentiation. Both *Tsix* pinpoints (arrowheads) and *Xist* clouds (asterisks) are visible. DNA is stained with DAPI (blue). Scale bar: 10 μm. **h** Quantification of (**g**), displaying the average percentage of nuclei with *Xist* clouds. Average percentage ± SD, the total number of counted nuclei per clone is indicated. **i** Relative allele-specific *Tsix* expression of Wt and Spen[−/−] ESCs at different time points of monolayer differentiation, determined by RT-qPCR. Average expression ± SD. Two-sided Mann–Whitney test (α < 0.05). Reference gene: *Hist2h2aa1*. (**c–f, h, i**) *n* = 4 biological replicates including 2 independent clones (squares vs. circles) per condition.

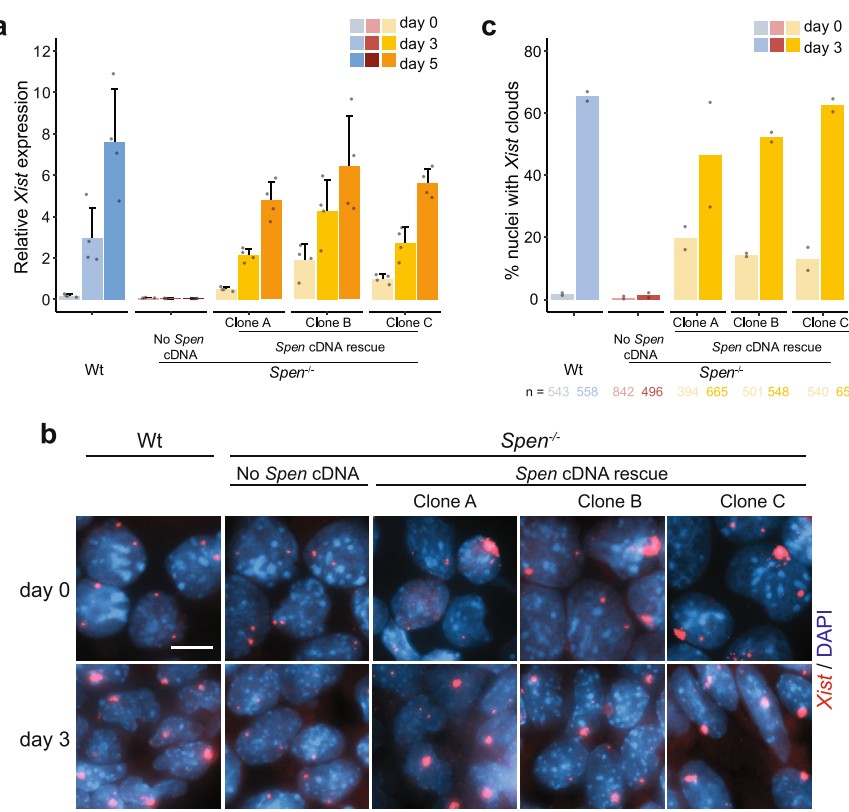

**Fig. 2 *Spen* cDNA expression in *Spen*[−/−] ESCs rescues *Xist* expression. a** Relative total *Xist* expression determined by RT-qPCR at days 0, 3, and 5 of monolayer differentiation of Wt, *Spen*[−/−] and three independent rescue clones (Clone A, B and C). Average expression ± SD, *n* = 4 biological replicates per condition. Reference gene: *Hist2h2aa1*. **b** RNA FISH of *Xist* (red) at days 0 and 3 of monolayer differentiation of Wt, *Spen*[−/−], and *Spen* rescue clones. DNA stained with DAPI (blue). Scale bar: 10 μm. **c** Quantification of (**b**), showing the percentage of nuclei with *Xist* clouds in each condition. Average percentage, *n* = 2 biological replicates, the total number of counted nuclei is indicated.

does not cause a differentiation phenotype. Interestingly, in undifferentiated *Spen*[−/−(cDNA)] ESC clones, *Xist* levels were higher than in Wt controls (Fig. 2a), and *Xist* RNA-FISH analysis revealed a significant percentage of *Spen*[−/−(cDNA)] undifferentiated ESCs having *Xist* clouds (Fig. 2b, c), probably due to higher SPEN abundance after overexpression in the undifferentiated state (Supplementary Fig. 3c).

**SPEN accumulation on the Xi early during XCI.** In light of the previous results, we expect SPEN to accumulate on the Xi at the very early steps of physiological XCI. To investigate this hypothesis and since commercial SPEN antibodies suitable for

immunofluorescence (IF) are lacking, we generated ESCs endogenously expressing a C-terminally tagged SPEN-GFP (Supplementary Fig. 4a). Correct GFP integration was confirmed by PCR on gDNA and by FACS analysis (Supplementary Fig. 4b,c). IF detection of GFP at different time points of monolayer differentiation revealed SPEN accumulation as early as day 1 in about 30% of the nuclei (Fig. 3a, b). This percentage remained stable until day 3, while from day 5 onward, SPEN accumulation increased to plateau at day 7. Interestingly, the intensity of SPEN-GFP accumulations gradually rises as differentiation progresses (Fig. 3c), suggesting that the number of SPEN molecules increases over time.

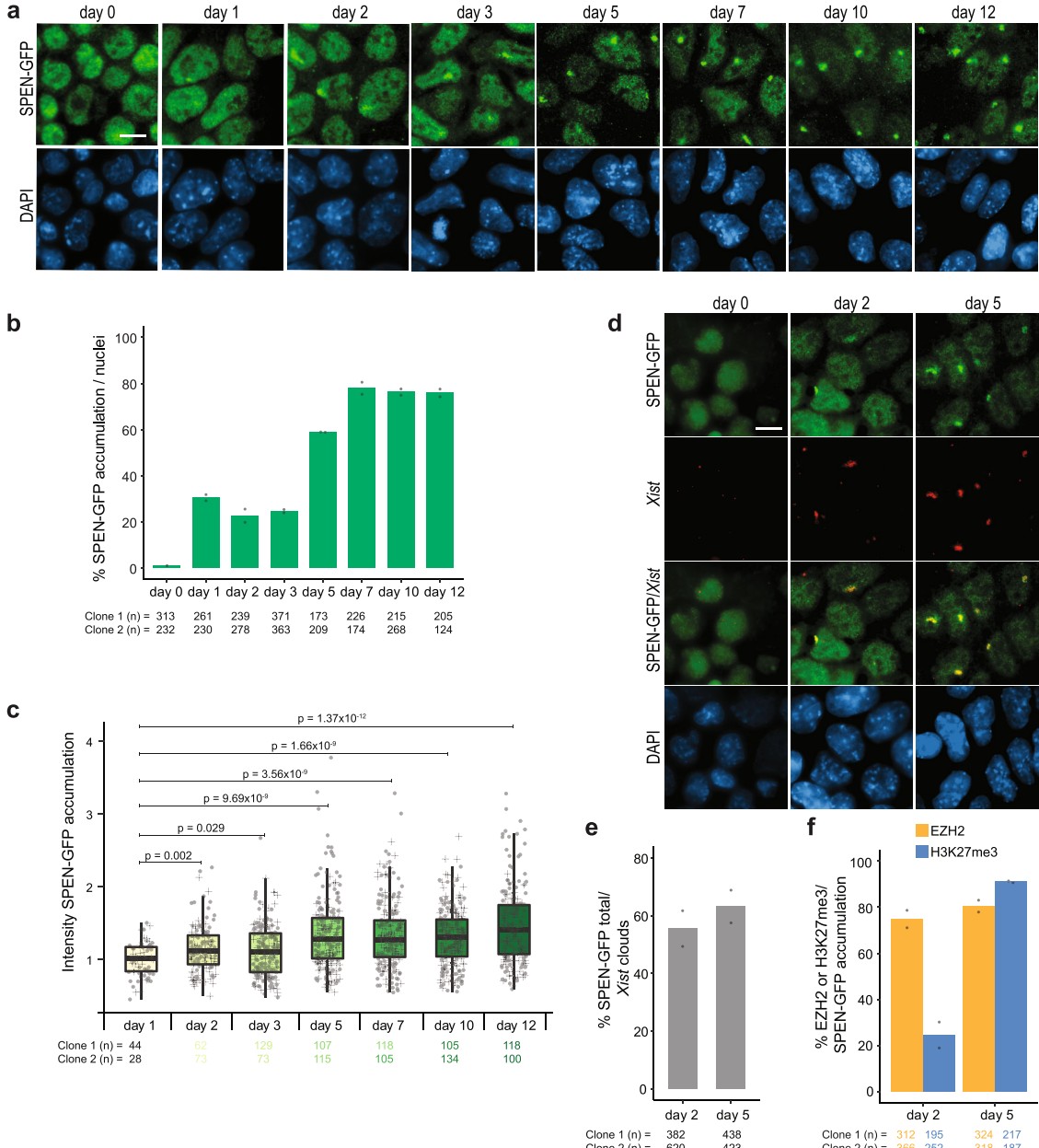

**Fig. 3 SPEN colocalizes with the Xi chromosome very early during monolayer differentiation. a** SPEN-GFP IF staining (αGFP, green) at different time points of monolayer differentiation (day 0, 1, 2, 3, 5, 7, 10, and 12). DNA is stained with DAPI (blue). Scale bar: 10 μm. **b** Quantification of (**a**), depicting the percentage of SPEN-GFP accumulation per nuclei at different time points of monolayer differentiation. Average percentage, $n = 2$ independent clones, the total number of counted nuclei per clone is indicated. **c** Boxplots illustrating the intensity of SPEN-GFP accumulations at different days of monolayer differentiation. The box plots display the median (central black line), the interquartile range (box limits) and the whiskers represent 1.5x of the interquartile range. All the data points are shown for two independent SPEN-GFP clones (filled circles vs. crosses). Two-sided Mann–Whitney test corrected with Benjamini–Hochberg for multiple testing ($\alpha < 0.05$). The total number of studied nuclei per clone is indicated. **d** IF (αGFP, green) combined with Xist RNA FISH (red) of SPEN-GFP lines at days 0, 2, and 5 of monolayer differentiation. DNA is stained with DAPI (blue). Scale bar: 10 μm. **e** Percentage of nuclei with a Xist cloud that also presents SPEN accumulation. Average percentage, $n = 2$ independent clones, the total number of counted nuclei per clone is indicated. Calculated from the same data as for Supplementary Fig. 4d. **f** Percentage of nuclei with SPEN accumulation that also present EZH2 or H3K27me3 accumulation. Average percentage, $n = 2$ independent clones, the total number of counted nuclei per clone is indicated. Calculated from the same data as for Supplementary Fig. 4g.

Previous studies have shown that SPEN and Xist colocalize in doxycycline-inducible Xist undifferentiated ESC lines[20,25], although the normal timing and relationship between SPEN and Xist in differentiating cells remained unexplored. Therefore, we investigated SPEN accumulation in relation to Xist, by performing GFP IF combined with Xist RNA FISH in differentiating cells (Fig. 3d, Supplementary Fig. 4d). This analysis

indicated that at day 2 of differentiation, about 55% of the cells with Xist clouds showed SPEN colocalization (Fig. 3e).

We also studied the relation of SPEN with other key players in XCI, including the PRC2 catalytic subunit EZH2 and its catalytic product H3K27me3. As found for Xist, we could detect colocalization of EZH2 and its associated histone modification with SPEN (Supplementary Fig. 4e–g). At day 2 of differentiation,

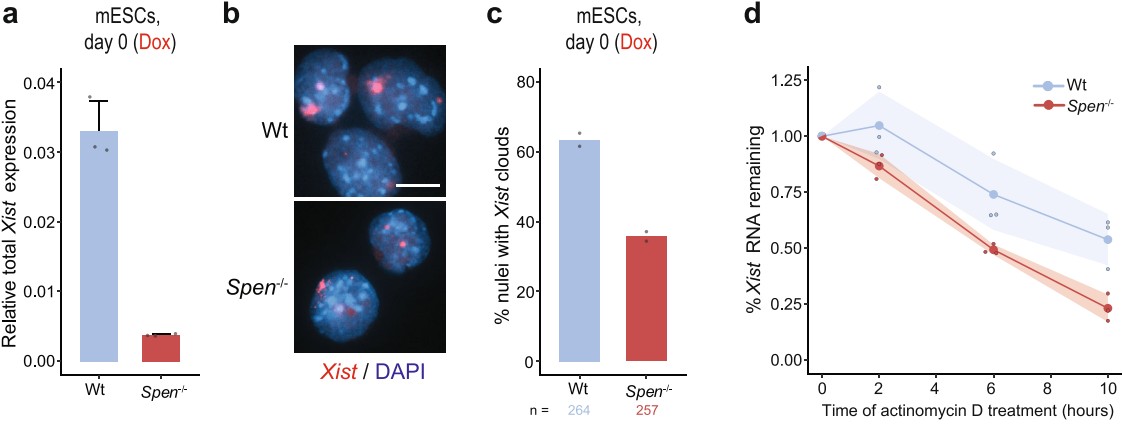

**Fig. 4 SPEN plays a role in *Xist* RNA stability. a** Relative total *Xist* expression in undifferentiated Wt and *Spen*$^{-/-}$ ESCs upon *Xist* induction with doxycycline for 4 days. *n* = 3 biological replicates per condition. Average expression ± SD. Reference gene: *Beta-actin*. **b** *Xist* RNA FISH (red) of undifferentiated Wt and *Spen*$^{-/-}$ ESCs treated with doxycycline to induce *Xist* expression. DNA is stained with DAPI (blue). Scale bar: 10 μm. **c** Quantification of (**b**), displaying the percentage of nuclei with *Xist* clouds. Average percentage, *n* = 2 biological replicates per condition, the total number of counted nuclei is indicated. **d** Percentage of *Xist* RNA remaining at different time points of actinomycin D treatment (*t* = 0, 2, 6 and 10 h) in undifferentiated Wt and *Spen*$^{-/-}$ ESCs treated with doxycycline. Normalized relative expression to *t* = 0 ± SD, shading represents the SD, *n* = 3 biological replicates per condition. Reference gene: *Beta-actin*.

about 70% of nuclei with SPEN accumulation displayed EZH2 and 20% H3K27me3 accumulation. At day 5, the percentage of nuclei with SPEN accumulation that also have EZH2 or H3K27me3 is of approximately 80% and 90%, respectively (Fig. 3f). Our results show that SPEN accumulation on the Xi happens very early upon differentiation in a progressive manner.

***Xist* RNA stability is compromised in *Spen*$^{-/-}$ ESCs.** We have shown that SPEN is required for endogenous *Xist* upregulation and XCI upon ESC differentiation. In addition, we noticed that forced *Xist* upregulation resulted in reduced *Xist* RNA levels in *Spen*$^{-/-}$ cells compared with Wt cells (Fig. 1d, Fig. 4a). Accordingly, the percentage of nuclei with *Xist* clouds was lower in *Spen*$^{-/-}$ cells with doxycycline-induced *Xist* versus Wt cells and *Xist* clouds were in general smaller (Fig. 4b, c), an effect that could be attributed to affected stability of *Xist*. To study the role of SPEN in *Xist* stability, we determined the half-life of doxycycline-induced *Xist* RNA in *Spen*$^{-/-}$ and Wt ESCs treated with actinomycin D to block its transcription. The remaining levels of *Xist* RNA at different time points were assessed by RT-qPCR, and the *Xist* RNA-decay rate and half-life were then calculated (Fig. 4d). In Wt ESCs, the *Xist* RNA half-life was 10 h 17 min, whereas the *Xist* half-life was reduced to 4 h 40 min in *Spen*$^{-/-}$ cells. These results indicate that SPEN plays a role in *Xist* RNA stability, but cannot explain why physiological *Xist* upregulation is lost in *Spen*$^{-/-}$ ESCs upon differentiation.

**SPEN, HDAC3, and H3K27ac are enriched at the *Tsix* regulatory region.** Our results showed that at late days of monolayer differentiation, *Spen*$^{-/-}$ cells display higher *Tsix* levels than control cells (Fig. 1i), suggesting that SPEN might be recruited by *Xist* to silence *Tsix*. Hence, we explored SPEN genomic binding at the *Xist–Tsix* locus using published SPEN CUT&RUN data in undifferentiated ESCs with a doxycycline-responsive *Xist* promoter[25]. SPEN accumulation is evident on the gene body of *Xist* as well as on the *Tsix* regulatory region at 24 h of doxycycline induction, but more prominently after 4 and 8 h of induction (Fig. 5a, b). This *Tsix* regulatory region comprises the minor and major *Tsix* promoters and *Xite*, an enhancer of *Tsix*[28]. *Xist*-mediated recruitment of SPEN is important for recruitment and/or activation of HDAC3, responsible for the removal of H3K27ac

from the future Xi[16,29]. Analysis of published HDAC3 and H3K27ac ChIP-seq data from undifferentiated female ESCs upon 24 h of *Xist* induction[29] reveals HDAC3 binding and H3K27ac loss at the *Tsix* regulatory region, where SPEN is recruited (Fig. 5a). Moreover, X-linked *Rnf12* (Supplementary Fig. 5a) and *Pgk1* (Supplementary Fig. 5b) display SPEN and HDAC3 enrichment at their promoters and H3K27ac loss upon *Xist* induction. Interestingly, *Rnf12*, an early silenced gene[30], shows SPEN and HDAC3 promoter binding in the undifferentiated state without *Xist* induction, suggesting that in the undifferentiated state, *Xist* might be sufficiently expressed at very low levels to partly silence *Rnf12*, explaining the *Rnf12* allelic ratio difference between Wt and *Spen*$^{-/-}$ ESCs at day 0 (Fig. 1e, Supplementary Fig. 3e).

To test whether SPEN recruitment leads to H3K27me3 accumulation, we performed H3K27me3 ChIP-seq on *Spen*$^{-/-}$ and Wt ESCs prior to and during differentiation (day 3) (Fig. 5a). This analysis showed that while Wt cells display H3K27me3 enrichment at the *Tsix*-regulatory region upon differentiation, *Spen*$^{-/-}$ cells do not, in accordance with their lack of *Xist* upregulation. Interestingly, the H3K27me3 hotspot, located at the 3′ end of *Tsix*[31,32], is clearly reduced upon differentiation both in Wt and *Spen*$^{-/-}$ cells, while Wt and *Spen*$^{-/-}$ ESCs show no difference in H3K27me3 levels, pointing to a SPEN-independent H3K27me3 enrichment at the hotspot (Fig. 5a). Altogether, these data support a model where *Xist*-mediated SPEN recruitment leads to *Tsix* promoter silencing.

**SPEN is required to silence *Tsix* to allow *Xist* upregulation.** To further investigate the role of SPEN in silencing *Tsix*, we generated compound *Spen*$^{-/-}$:*Tsix*-defective hybrid ESC lines. If *Xist*-mediated recruitment of SPEN to the *Tsix* regulatory region is crucial for *Tsix* silencing and *Xist* upregulation, we expect *Xist* upregulation upon differentiation to be rescued in these double-knockout cell lines. We made use of a *Tsix*-Stop line containing a triple poly(A) signal downstream of the major *Tsix* promoter that blocks its transcription on the 129 allele[7], and a *Tsix*-Cherry line with a mCherry coding sequence introduced downstream of the major *Tsix* promoter on the Cast allele[33] (Fig. 6a). As controls, we generated *Spen*$^{-/-}$ ESCs in the same hybrid background (F1:129/Cast) where both *Xist* alleles are intact[34], in contrast to the previously studied ESCs that contained a doxycycline-inducible *Xist*

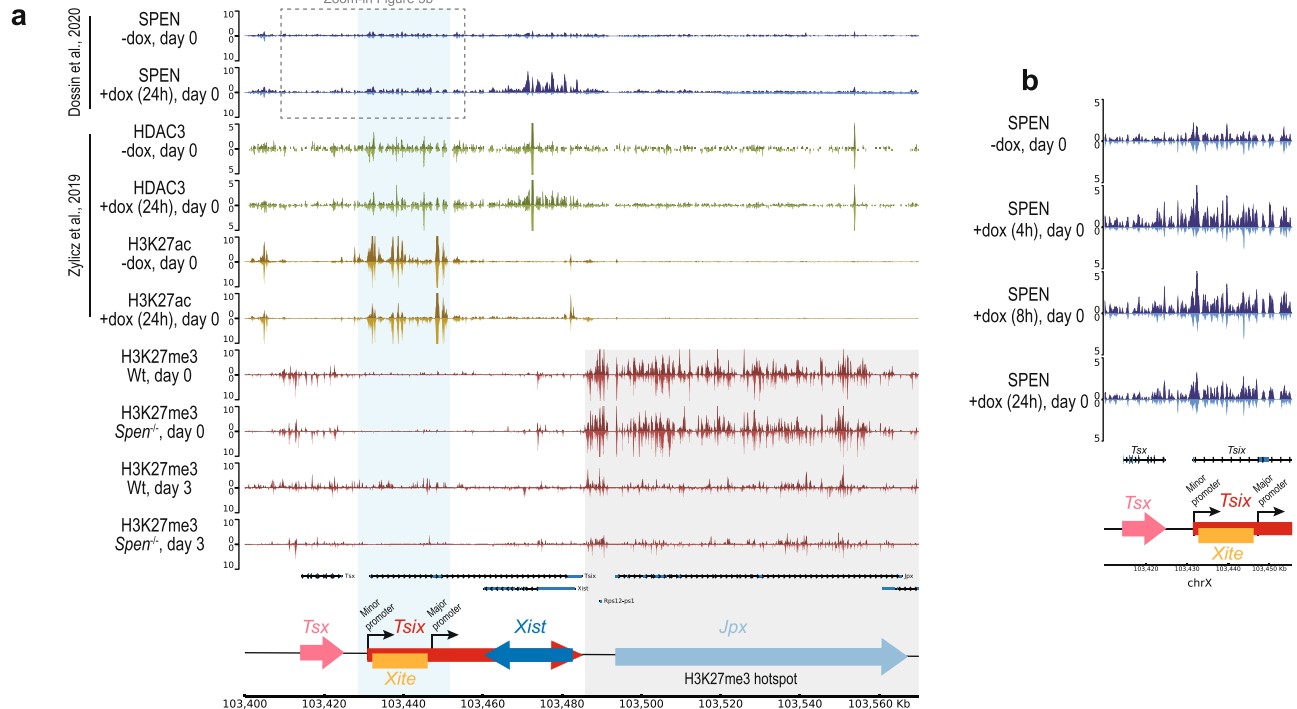

**Fig. 5 *Tsix*-regulatory region shows SPEN, HDAC3 and H3K27ac enrichment. a** Genome browser tracks showing allele-specific SPEN, HDAC3, H3K27ac and H3K27me3 binding at the *Xist–Tsix* locus. The top part (dark) of each track represents the Xi and the bottom (light) the Xa. SPEN CUT&RUN (blue, top) profile in *Xist*-inducible undifferentiated ESCs (day 0) untreated or treated with doxycycline (24 h)[25]. HDAC3 (green, middle–top) and H3K27ac ChIP-seq (yellow, middle–bottom) in *Xist*-inducible undifferentiated ESC (day 0) untreated or treated with doxycycline (24 h)[29]. H3K27me3 ChIP-seq (red, bottom) in Wt and *Spen*$^{-/-}$ ESCs at days 0 and 3 of monolayer differentiation. The light-blue rectangle highlights the *Tsix* regulatory region, including the *Tsix* minor and major promoters, and *Xite*. The light-gray rectangle highlights the H3K27me3 hotspot region. The dashed rectangle indicates the genomic region in (b). **b** Zoom-in view of the *Tsix* promoter region, showing the SPEN CUT&RUN profile in undifferentiated ESCs (day 0) untreated or treated (4 h, 8 h and 24 h) with doxycycline.

allele. Deletion of the *Spen* ORF was confirmed in all cell lines by PCR on gDNA and Western blot analysis (Supplementary Fig. 6a,b). We generated two independent knockout clones per line. All ESC lines were differentiated in biological quadruplicates, followed by allele-specific *Xist* RNA expression analysis by RT-qPCR. As expected, the F1:129/Cast control *Spen*$^{-/-}$ line showed no *Xist* upregulation (Fig. 6b), similar to the phenotype observed in the *Spen* knockout clones generated in the heterozygous doxycycline-inducible *Xist* cell line (Fig. 1f). Remarkably, the introduction of a poly(A) signal in *Spen*$^{-/-}$:*Tsix*-Stop lines fully rescued *Xist* upregulation upon differentiation (Fig. 6c), indicating that SPEN is required for *Xist* upregulation via *Tsix* repression. However, the *Spen*$^{-/-}$:*Tsix*-Cherry lines displayed only mild upregulation of *Xist* upon differentiation (Fig. 6c). These differences in the rescue phenotype between *Spen*$^{-/-}$:*Tsix*-Cherry and *Spen*$^{-/-}$:*Tsix*-Stop lines can be explained by the remaining *Tsix* transcription in the *Tsix*-Cherry line, which is fully ablated in the *Tsix*-Stop lines (Supplementary Fig. 6c). As expected, the *Spen*$^{-/-}$: *Tsix*-Stop lines are not able to silence *Rnf12*, despite normal *Xist* levels in the *Tsix*-Stop line (Supplementary Fig. 6d). Moreover, the Spen$^{-/-}$:*Tsix*-Stop lines display *Xist* clouds that were similar in morphology and number compared with Wt F1:129/Cast cells (Fig. 6d, e).

Using a doxycycline-inducible *Xist* line, we have shown SPEN's importance in *Xist* RNA stability (Fig. 4d). However, since doxycycline-induced *Xist* expression is about 10 times higher compared with its endogenous expression upon differentiation (Fig. 1d,f), we wanted to analyze the SPEN's role in *Xist* RNA stability when expressed endogenously. For this, we used Wt: and *Spen*$^{-/-}$:*Tsix*-Stop ESCs at day 3 of monolayer differentiation,

where *Xist* is upregulated from its endogenous promoter (Fig. 6c). In Wt:*Tsix*-Stop ESCs, *Xist* RNA's half-life was 9 h 42 min, while in Spen$^{-/-}$:*Tsix*-Stop ESCs, it was reduced to 2 h 46 min (Fig. 6f), confirming that SPEN is important for *Xist* RNA stability, independently of the promoter type and RNA abundance.

Taken together, our results indicate an essential role for SPEN in *Xist* upregulation, mainly via silencing of *Tsix*, and to a lesser extent by stabilizing *Xist* RNA. SPEN is therefore not only crucial in X-linked gene silencing but also in the early initiation steps of XCI, playing a role in the feedforward loop leading to *Xist* activation (Fig. 7).

**Discussion**

In this study, we show that SPEN-defective ESCs do not upregulate *Xist* upon differentiation and study the molecular mechanism behind this observation. To explore the role of SPEN in XCI, various studies used *Xist*-inducible ESC lines to generate SPEN-knockdown or -knockout cell lines (Supplementary Table 1). Forcing *Xist* expression in ESCs is a powerful way to understand X-linked gene silencing, but is not suitable to investigate the initiation phase of XCI. Studies exploring the role of SPEN in cells undergoing physiological XCI involved *Spen* knockdown strategies[17,20], therefore, the levels of SPEN might have been sufficient to allow normal *Xist* upregulation, while showing a defect in X-linked gene silencing. Likewise, our *Spen*$^{+/-}$ ESC lines show *Xist* upregulation, but reduced X-linked silencing compared with Wt lines. In agreement with our results, one study reports lower *Xist* abundance and cloud formation upon forced *Xist* induction from the endogenous locus in *Spen*$^{-/-}$ ESCs[27]. Another recent study suggested that *Spen*$^{-/-}$ ESCs are not able to

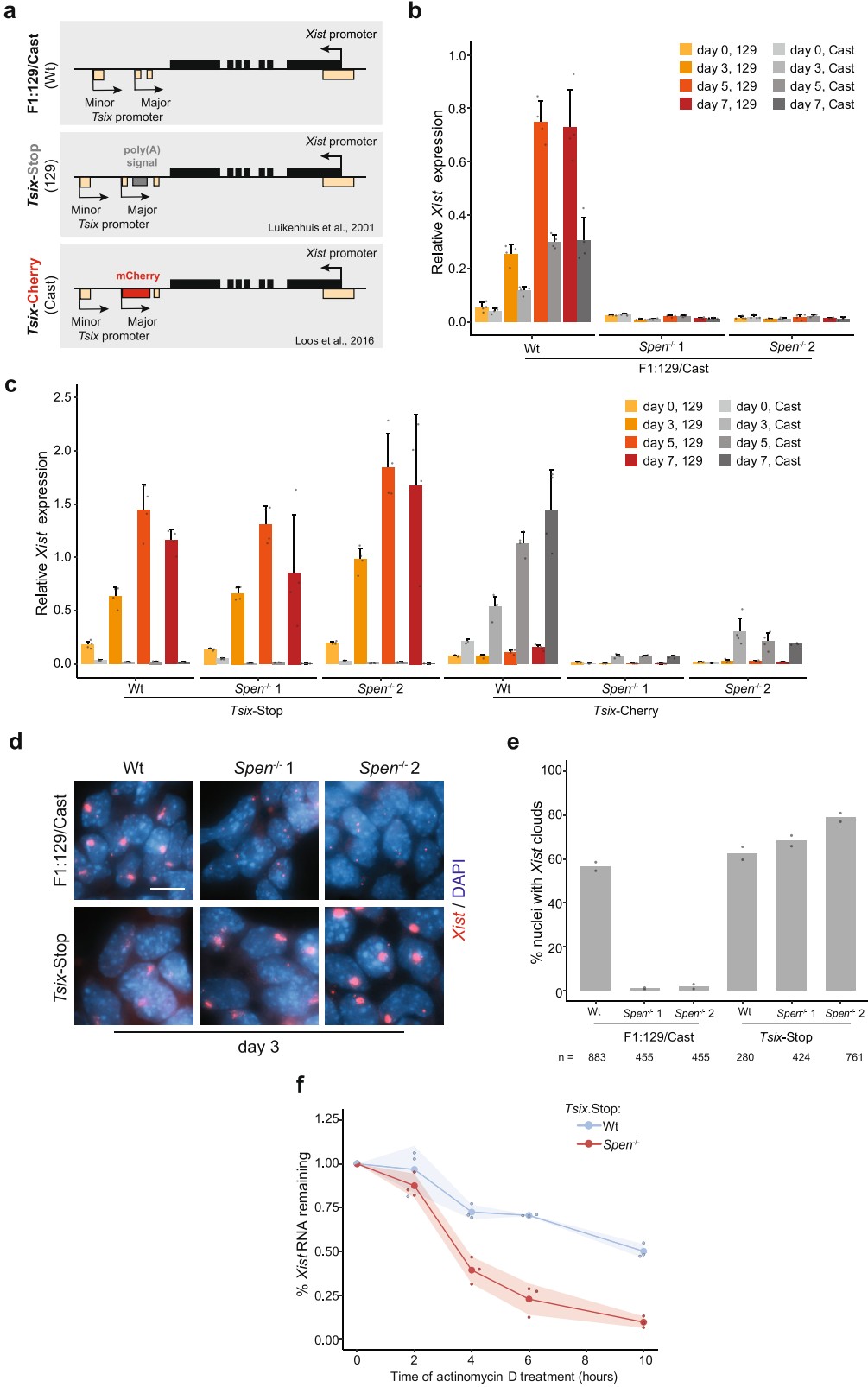

differentiate upon leukemia inhibitory factor (LIF) removal and differentiation towards neural progenitor cells[35], while we observe that $Spen^{-/-}$ ESCs display a normal morphology in the undifferentiated state and undergo normal differentiation upon monolayer differentiation. Nevertheless, we observe more cell death of $Spen^{-/-}$ cells compared with Wt cells upon monolayer differentiation, although we consider that this might not be XCI-

related since X0 $Spen^{-/-}$ cells also die upon differentiation (data not shown). This observation is probably not surprising since SPEN plays a role in various biological processes as a transcriptional repressor[23,24,36].

Previous studies performed with doxycycline-inducible $Xist$ lines show that SPEN and $Xist$ colocalize soon after $Xist$ induction[20,25]. However, $Xist$ upregulation from a doxycycline-

**Fig. 6 SPEN is required to silence *Tsix*, to allow *Xist* upregulation upon XCI initiation. a** Schematic overview of the *Tsix*-defective ESC lines used to study the role of SPEN in *Tsix* silencing and *Xist* upregulation, namely, Wt (F1:129/Cast) (top), *Tsix*-Stop (middle), and *Tsix*-Cherry (bottom). The *Tsix*-Stop line is defective for *Tsix* in the 129 allele; the *Tsix*-Cherry line is defective in the Cast allele. Homozygous deletion of *Spen* was performed in the three lines. **b** Relative allele-specific *Xist* expression in Wt and *Spen*[−/−] F1:129/Cast ESC lines at different time points of monolayer differentiation. Average expression ± SD, n = 4 biological replicates per condition. Reference gene: *Hist2h2aa1*. **c** Same as in (**b**) for Wt and *Spen*[−/−] *Tsix*-Stop and *Tsix*-Cherry ESC lines. **d** *Xist* RNA FISH (red) at day 3 of monolayer differentiation in F1:129/Cast and *Tsix*-Stop Wt and *Spen*[−/−] ESC lines. DNA is stained with DAPI (blue). Scale bar: 10 μm. **e** Quantification of (**d**), showing the percentage of nuclei with *Xist* clouds. Average percentage, n = 2 biological replicates per condition, the total number of counted nuclei is indicated. **f** Percentage of *Xist* RNA remaining at different time points of actinomycin D treatment (t = 0, 2, 4, 6, and 10 h) in differentiating Wt and *Spen*[−/−] *Tsix*-Stop ESCs (day 3). Normalized relative expression to t = 0 ± SD, shading represents the SD, n = 3 biological replicates. Reference gene: *Beta-actin*.

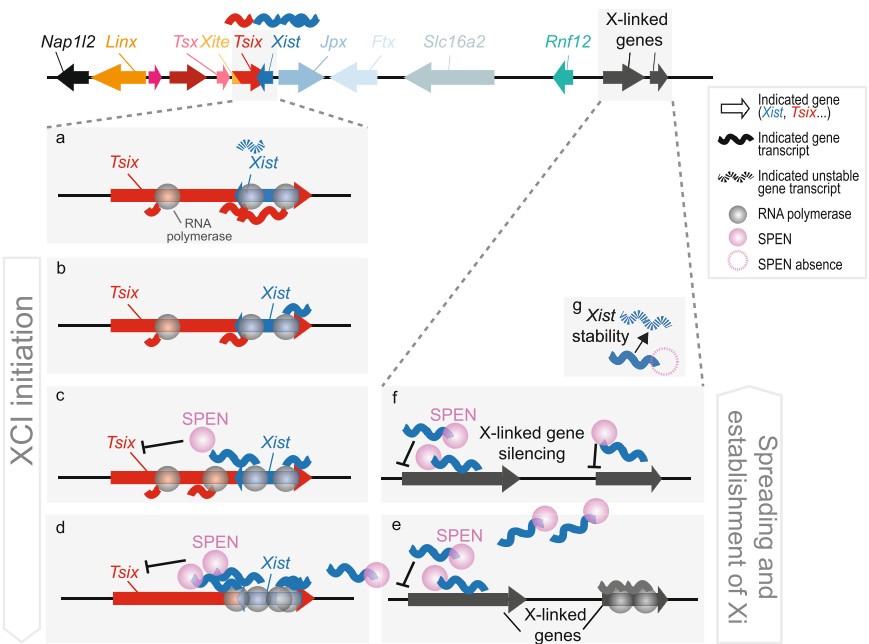

**Fig. 7 Model of the role of SPEN in XCI.** In the undifferentiated state, *Tsix* transcription represses *Xist* and *Tsix* RNA levels are high (**a**). Upon differentiation, at the start of XCI (**b**–**d**), *Xist* transcription increases (**b**), the nascent *Xist* transcripts recruit SPEN that is necessary to silence the *Tsix* promoter through removal of active H3K27ac marks (**c**). Consequently, the silencing of *Tsix* allows *Xist* upregulation and accumulation (**d**). Later, during the establishment phase of XCI, the spreading of SPEN-*Xist* along the X chromosome allows the silencing of active genes (**e**, **f**). Moreover, SPEN is also necessary to stabilize *Xist* RNA (**g**). This model, in addition to previous evidence, proposes that SPEN is not only key in X-linked gene silencing, but also in initiation of XCI.

inducible promoter happens at a very different timescale than during physiological XCI. In the present study, we observe that cells with SPEN accumulation are detectable from day 1 of differentiation with their number and intensity levels of these accumulations increasing over time, suggesting that SPEN accumulates in a progressive manner.

In addition, we provide evidence that SPEN is required to silence *Tsix* to allow *Xist* upregulation. (I) In *Spen*[−/−] cells, we detect higher levels of *Tsix* in differentiating cells from the allele undergoing X-chromosome inactivation, compared with Wt cells. In line with this, paternal *Tsix* levels in female E3.5 blastocysts with a mutated *Xist* RepA, necessary for SPEN recruitment, are higher than in Wt blastocysts[37], which is suggestive of defective *Tsix* silencing in the absence of SPEN. (II) SPEN overexpression in ESCs leads to higher *Xist* levels and ectopic *Xist* cloud formation in the undifferentiated state, suggesting that SPEN overexpression might lead to partial silencing of the *Tsix* promoter, facilitating *Xist* expression. (III) SPEN binds to active promoters and enhancers to silence the X chromosome[25]. Spatial proximity to the *Xist* locus is a strong predictor of X-linked gene silencing efficiency[26,38]. One of the closest actively transcribed promoters to *Xist* is *Tsix* and we indeed observe SPEN binding at the *Tsix* regulatory region. (IV) *Spen*[−/−] cell lines

cannot upregulate *Xist*, while compound *Spen*:*Tsix* defective cell lines (*Spen*[−/−]:*Tsix*-Stop) display normal *Xist* levels upon differentiation, indicating that in the absence of SPEN, *Tsix*, which acts as a brake on *Xist* transcription, cannot be silenced. Furthermore, our results show that SPEN plays a role in *Xist* stability, both upon forced *Xist* upregulation in undifferentiated ESCs and in differentiating *Tsix*-Stop ESCs, in line with a recently published study, where a role for SPEN in RNA *Xist* stability is described using microscopy-based approaches[39].

During initiation of XCI, when *Xist* starts to be transcribed from the future Xi, SPEN helps remodel the chromatin environment of the *Xist*–*Tsix* locus. SPEN and HDAC3 are present at the *Tsix* regulatory region and H3K27ac levels decrease upon *Xist* induction. Accordingly, we propose that during XCI initiation, SPEN binds *Xist* nascent transcripts[25], recruits and/or activates HDAC3[16,29], which removes histone acetylation marks, weakening the *Tsix* promoter activity and facilitating *Xist* expression. Then, *Xist* spreads together with SPEN along the X-chromosome to silence active genes. Moreover, SPEN is necessary for *Xist* RNA stabilization (Fig. 7).

The mechanism driving the symmetry breaking event leading to monoallelic upregulation of *Xist* has been a focus of many

 

studies. Several studies showed that the *Xic* and more specifically the *Xist-Tsix* master switch are tightly regulated in a deterministic process, involving monoallelic downregulation of *Tsix* or X-pairing mechanisms[40–42]. In contrast, our studies, and studies of others, indicate that a gene regulatory network composed of transacting activators and inhibitors of XCI, involving positive- and negative-feedback loops, instructs the *cis*-regulatory landscape of the *Xic* to direct monoallelic upregulation of *Xist*[12,34,43,44]. *Tsix* silencing not only involves downregulation of inhibitors of XCI, including pluripotency factors OCT4, SOX2, KLF4, c-MYC, and REX1[45–47], but also involves *Xist*-mediated recruitment of SPEN as our present work reveals. This can happen on either allele, but asynchronous *Xist* transcription bursts will facilitate *Xist*-mediated monoallelic silencing of *Tsix* through SPEN, resulting in further upregulation of *Xist* and concomitant silencing of the XCI activator *Rnf12*, providing a negative feedback loop to prevent upregulation of *Xist* on the future active X chromosome. In humans, SPEN seems to play a role in XCI[48] even though in-depth mechanistic studies are missing, but whether SPEN is required for *XIST* upregulation is unknown. In addition, *TSIX* does not completely overlap *XIST* and does not seem to be a strong regulator of *XIST*[49,50], therefore, a mechanism in which SPEN negatively regulates *TSIX* to allow *XIST* upregulation is not very likely. However, the possibility that SPEN controls other human-specific *XIST* regulators should be taken into consideration.

## Methods

**Cell culture**. Mouse ESCs were grown on feeder cells, Wt C57BL/6 male-irradiated mouse embryonic fibroblasts (MEFs), and medium containing DMEM (Gibco, 11995065), 15% fetal calf serum (FCS) (Capricorn Scientific, FBS-12A), 0.1 mM nonessential amino acids (NEAA) (Lonza, BE13-114E), 100 U mL$^{-1}$ penicillin, 100 ug mL$^{-1}$ streptomycin (Sigma-Aldrich, P0781), 0.1 mM 2-mercaptoethanol (Gibco, 31350010) and 5.000 U mL$^{-1}$ LIF. Previous to monolayer differentiation cells were plated in nongelatinized plates to eliminate feeder cells. Then, plated at specific densities per time point in differentiation medium composed of IMDM-glutamax (Gibco, 31980030), 15% FCS, 0.1 mM NEAA, 100 U mL$^{-1}$ penicillin, 100 µg mL$^{-1}$ streptomycin, 37.8 µL L$^{-1}$ monothioglycerol (Sigma-Aldrich, M6145) and 50 µg mL$^{-1}$ ascorbic acid (Sigma-Aldrich, A92902). When appropriate, the medium was supplemented with 2 µg mL$^{-1}$ doxycycline (Sigma-Aldrich, D9891).

**Gene editing using the CRISPR/Cas9 technology**. Different female F1 2–1 hybrid (129/Sv-Cast/Ei) ESC lines with different genetic modifications to interrogate various aspects of XCI were targeted in this study (Supplementary Table 2). To generate *Spen* heterozygous and homozygous knockout clones two single-guide RNAs (sgRNA) targeting the 5′ (5′-AGTGCGCTTCGTCACTGCAC-3′) and 3′ (5′-TCCTCCCGCCCCGACGCGGA-3′) region of the *Spen* ORF were cloned in the Cas9-GFP pX458 vector (Addgene plasmid #48138). Compatible 5′ and 3′ homology arms (HA) of approximately 500 bp were amplified by PCR from mouse gDNA and cloned in the pCR-Blunt-II-TOPO vector with a NdeI restriction site in-between the 5′ and 3′ HA. This site was used to insert a puromycin resistance (PuroR) cassette flanked by loxP sites. The *ROSA26* locus was targeted using the pX458 vector coding for a sgRNA (5′-CGCCCATCTTCTAGAAAGAC-3′) compatible with the pFD46 expression vector[25], coding for the *Spen* cDNA and a hygromycin resistance cassette. To create an endogenous *Spen* C-terminal enhanced GFP (eGFP) knock-in, a sgRNA targeting the 3′ end of *Spen* ORF (5′-ATTGTCATTGCCTCGGTGTG-3′) was cloned in the Cas9-PuroR pX459 vector (Addgene plasmid #62988). The donor template was made using a gblock from Integrated DNA Technologies coding for compatible 5′ and 3′ HA of 600 bp with a NheI and AscI restrictions sites in-between the 5′ and 3′ HA, which were used to insert an eGFP in frame with the *Spen* coding sequence. To clone the sgRNAs, two complementary oligonucleotides coding for the desired target sequence and overhangs compatible with the restriction enzyme BsbI were annealed and ligated in the pX458 or px459 plasmids, that were previously digested with BsbI. The appropriate plasmid combinations were transfected into ESCs using Lipofectamine 2.000 (Invitrogen, 11668019) and plated at low density to obtain single colonies, when the appropriate medium was supplemented with 1 µg mL$^{-1}$ puromycin (Sigma-Aldrich, P8833) for 48–72 h or 250 µg mL$^{-1}$ hygromycin B (Invitrogen, 10687010) for 7 days. Colonies were screened by PCR for correct integration of the desired construct (Supplementary Table 3). Positive clones were further characterized by Western blot, RT-qPCR and/or FACS, and the presence of 2 X chromosomes and correct karyotype was also assessed. Full scan images of all the genotyping gels are available in the Source Data file.

**Protein extraction and Western blot**. To prepare nuclear extracts all procedures were done at 4ºC and buffers supplemented with 1x protease inhibitors (Roche, 4693132001), 15 µM MG-132 (Sigma-Aldrich, C2211) and 0.5 mM DTT. Cells were harvested by scraping in cold PBS, collected and centrifuged (200 × g, 5 min, 4°C). The pellet was incubated in 5x the pellet volume of Buffer A (10 mM Hepes pH 7.6, 1.5 mM MgCl$_2$, and 10 mM KCl) for 10 min, vortexed (30 sec) and centrifuged (900 × g, 5 min, 4°C). Then, the pellet was resuspended in 1.5x Buffer C (20 mM Hepes pH 7.6, 25% glycerol, 420 mM NaCl, 1.5 mM MgCl$_2$, and 0.2 mM EDTA) and rotated (30 min, 4°C). The solution was centrifuged (18,000 × g, 10 min, 4°C) and the supernatant collected as nuclear extract. The protein concentration was measured using Nanodrop and all samples diluted to the same concentration using Buffer C. NuPAGE™ LDS Sample Buffer (4X) (Invitrogen, NP0007) containing 5% b-mercaptoethanol was added to nuclear extracts, and boiled (95ºC, 5 min). For Western blot analysis the NuPAGE™ 3–8% Tris-Acetate gels (Invitrogen, EA03755) were used with Tris-Acetate SDS Running Buffer (pH 8.24) (Invitrogen, LA0041) and the HiMark Pre-Stained Protein Standard (Invitrogen, LC5699). Wet transfer on a PVDF membrane was done overnight at 40 mA, with the NuPAGE™ Transfer Buffer (Invitrogen, NP00061), containing 10% methanol and 0.1% SDS. After blocking, the membrane was incubated with the appropriate primary antibodies, SPEN antibody (Abcam, ab72266, 1:2.000), Anti-Flag antibody M2-Peroxidase (Sigma-Aldrich, A8592, 1:3.000) and/or VCP antibody (Abcam, ab11433, 1:20.000), and appropriate secondary antibodies, IRDye 800CW donkey-anti-rabbit (LI-COR Biosciences, P/N 925-32213, 1:10.000), IRDye 680RD donkey-anti-mouse (LI-COR Biosciences, P/N 925-68072, 1:15.000) and/or Goat anti-mouse–Peroxidase antibody (Sigma-Aldrich, A4416, 1:4.000). Full scan images of all the Western blots are available in the Source Data file.

**RT-qPCR**. Total RNA was isolated from cell pellets using the ReliaPrep RNA Cell Miniprep System (Promega, Z6012) and reversed-transcribed using Superscript III (Invitrogen, 18080093) and random hexamers (Invitrogen, N8080127), following the manufacturer's instructions. All RT-qPCRs were done using the GoTaq qPCR Master Mix (Promega, A6002) in a CFX384 real-time PCR-detection system (Bio-Rad). *Hist2h2aa1* was used as a normalization control, except in the RNA-stability assays where *Beta-actin* was used. All expression primers are listed in Supplementary Table 4. The optimal allele-specific primer-pair concentration to amplify the 129 and Cast allele at the same efficiency was optimized using pure 129, Cast and 129-Cast gDNA.

**RNA-stability assay**. To study the *Xist* RNA stability in undifferentiated ESCs, Wt and *Spen*$^{-/-}$ ESC lines with a doxycycline-responsive *Xist* promoter were cultured in ESC medium supplemented with doxycycline to induce *Xist* expression for four days. While in doxycycline treatment, 5 µg mL$^{-1}$ actinomycin D (Sigma-Aldrich, A1410) was added for different time ($t = 0, 2, 6$, and 10 h). Before collection, ESCs were plated in nongelatinized plates to remove the feeders. To study the *Xist* RNA stability in differentiating Wt and *Spen*$^{-/-}$ *Tsix*-Stop ESCs, cells were differentiated, as explained in the "Cell culture" methods section, and 5 µg mL$^{-1}$ actinomycin D were added for different time ($t = 0, 2, 4, 6$, and 10 h). In both cases, cells were harvested for RNA isolation, cDNA synthesis and RT-qPCR. Then, total *Xist* levels were determined and normalized to *Beta-actin* and the percentage of the remaining *Xist* RNA was calculated by dividing *Xist* expression levels at the different times of collection relative to $t = 0$. Using linear regression analysis the RNA decay rate constant ($k_{decay}$) was calculated from the slope of the curve that best fitted our data and the RNA half-life ($t_{1/2}$) obtained with the following formula $t_{1/2} = \frac{ln2}{k\,decay}$.

**Immunofluorescence (IF)**. Undifferentiated ESCs were attached to slides using a cytospin, while differentiating cells were grown on coverslips coated with gelatin (Sigma-Aldrich, G1890) or laminin (Sigma-Aldrich, L2020). Cells were fixed with 4% paraformaldehyde (PFA) in PBS (10 min, RT), permeabilized with 0.4% Triton-X100 and 5% goat serum in PBS (15 min, on ice) and blocked with 10% goat serum in PBST, composed of 0.05% Tween-20 in PBS (30 min, RT). Primary antibody incubation was done in blocking buffer (2 h, RT), after three washings with PBST the slides were incubated with the secondary antibody (1 h, RT). After three washings, the second washing containing DAPI, the slides were mounted with ProLong Gold antifade mounting medium (Invitrogen, P36930). The used primary antibodies were the following: αGFP (Abcam, ab290, 1:500), H3K27me2/3 (Active motif, 39535, 1:100) and EZH2 (BD43 clone, kindly provided by Dr. Kristian Helin, 1:100). The used secondary antibodies were Goat-anti-Rabbit-488 (Invitrogen, A-11008 1:1.000) and Goat-anti-Mouse-546, (Invitrogen, A-1103, 1:400). Images were taken using a ZEISS Axio Imager M2 using digital microscopy camera AxioCam 503 and analyzed with ImageJ. To quantify the intensity of the SPEN-GFP accumulation at different days of monolayer differentiation, we determined the mean intensity of a region of interest (ROI) covering a specific SPEN-GFP accumulation. Next, this value was normalized to the mean nuclei background intensity excluding the SPEN-GFP ROI, calculated as the average intensity of 10 different nuclei per image.

**IF combined with RNA fluorescent in situ hybridization (FISH)**. The *Xist* FISH probe was made from 2 µg of a 5.5 kb DNA fragment of mouse *Xist* comprising

exons 3–7. Fluorescent labeling was done with dUTP SpectrumRed using the nick translation kit (Abbott, 07J00-001) overnight at 16ºC. The probe was purified with ProbeQuant G-50 Micro Columns (GE, GE28-9034-08), combined with 100 μg of mouse tRNA, 20 μg of mouse Cot-1 DNA (Invitrogen, 18440-016) and 100 μg of salmon sperm DNA (Invitrogen, 15632-011) and precipitated with 2 M NaAc (pH 5.6) and 100% EtOH. The pellet was resuspended in 50 μL hybridization mix (50% formamide, 10% dextran sulfate in 2xSSC) and stored at −20°C. Before use, 20 μL of probe was prehybridized (10 min, 75ºC) with 0.5 μg of mouse Cot-1 DNA supplemented with 10 mM vanadyl ribonucleoside complex (VRC) (NEB, S1402S) and 0.2 U μL⁻¹ RNAseOUT (Invitrogen, 10777019).

Cells on coverslips or slides were fixed with 4% PFA in PBS (10 min, RT), permeabilized with 0.5% Triton-X100 in PBS and blocked in TS-BSA buffer (0.1 M Tris-HCl (pH 7.5), 0.15 M NaCl, and 2 mg mL⁻¹ BSA (Jena Bioscience, BU-102) in H₂O). The αGFP (Abcam, ab290, 1:500) primary antibody was incubated in blocking buffer (30 min, 37°C), washed in PBS (3x) and then incubated with the secondary antibody (30 min, 37°C). All solutions were supplemented with 10 μM VRC and 0.2 U μL⁻¹ RNAseOUT. Slides were washed in PBS (3x), post-fixed with 4% PFA (10 min, RT), and washed again in PBS (x3). Dehydration was done with increasing ethanol concentrations (70%, 90% and 100%). The prehybridized probe was added on top of the coverslips or slides (20 h, 37ºC, humid chamber). Various washings were done with 50% formamide/2x SSC (5 min, 37ºC, 2x), 2x SSC (5 min, 37ºC, 2x) and TS buffers (5 min, RT, 2x). Then the slides were mounted and visualized as explained in the "IF" section.

**FISH**. The FISH only protocol was done as described in the IF-FISH section, skipping the IF section. Directly after the permeabilization, the dehydration step was performed.

**RNA-seq**. Doxycycline treated cells were sorted to isolate dsRed positive cells upregulating *Xist* from the desired allele. RNA was isolated using the ReliaPrep RNA Cell Miniprep System. A total of 16 DNA libraries were created, a maximum of 1 ng of total RNA was used following the Smart-seq2 protocol[51] without modifications. Then, we used the Nextera DNA Flex library prep kit (Illumina) to create a library from full-length cDNA. Samples were sequenced on a HiSeq2500 sequencer (50 bp single-end reads).

**Chromatin immunoprecipitation (ChIP)-seq**. In total, $50 \times 10^6$ cells were trypsinized, resuspended and fixed in 50 mL of warm medium and 1% PFA for 10 min at 37ºC. About 2.5 mL of Glycine 2.5 M were added to the cells (final concentration 0.125 M) to quench the PFA, 5 min RT on a rotator. All buffers from now on contain protease inhibitors (Roche, 4693132001). Cells were washed twice with cold PBS. Then 1x in 10 mL of Buffer 1 (10 mM Hepes pH 7.5, 10 mM EDTA, 0.5 mM EGTA, and 0.75% Triton X-100) and 1x in Buffer 2 (10 mM Hepes pH 7.5, 200 mM NaCl, 1 mM EDTA, 0.5 mM EGTA), 10 min rotating at 4ºC. Nuclei were then resuspended in Lysis/Sonication Buffer (150 mM NaCl, 25 mM Tris-HCl pH 7.5, 5 mM EDTA, 1% Triton, 0.1% SDS, 0.5% Sodium deoxycholate) and incubated for 30 min on ice. Nuclei were then sonicated 2 × 15 min (30″ ON/OFF, max input, ice cold water) in a Bioruptor. A small fraction of the lysate was run on a gel to confirm size population of 100–500 bp. About 1 μg of H3K27me3 antibody (Cell Signaling, 9733 S) was conjugated with 25 μL of magnetic beads (Life Technologies, 10004D) for 3 h rotating at 4ºC. About 25 ug of chromatin was IP'd with the Ab-beads overnight rotating at 4ºC. Beads were washed 2x in standard RIPA buffer (140 mM NaCl, 10 mM Tris-HCl pH 7.5150 mM NaCl, 1 mM EDTA pH8.0, 0.5 mM EGTA pH 8.0, 1% Triton, 0.1% SDS, 0.5% Sodium deoxycholate), and 1x in High Salt RIPA (same as standard RIPA but with 500 mM NaCl), 1x LiCl RIPA (same as standard RIPA but with 250 mM LiCl instead of NaCl) and rinsed once with TE, 10 min 4ºC each wash. Chromatin was eluted with 450 μL of Elution Buffer (1% SDS; 0.1 M NaHCO₃ in H₂O) with 22 μL protease K (10 mg mL⁻¹) and 5 μL RNAse A (10 mg mL⁻¹) and shaken at 1,000 rpm on an orbital thermal-mixer for 2 h at 37°C first and then 65ºC overnight. DNA was then Phenol–Chloroform extracted and resuspended in 20 μL of H₂O. About 0.25 μL was used per PCR to confirm the ChIP's success. The concentration was then measured and a ChIP-seq library was prepared using the ThruPLEX DNA-Seq Kit (Takara Bio, R400675), by following the manufacturer's instructions: template preparation to obtain a homogeneous double-stranded DNA material, library synthesis to ligate stem-loop adapters and a final library amplification step to extend the template, remove the stem-loop adapters and amplify the library for posterior sequencing on an Illumina HiSeq 2500 sequencer (50 bp paired-end reads).

**NGS data analysis: allele-specific RNA-seq and ChIP-seq**. Both the RNA-seq and the ChIP-seq data were processed allele-specifically. The single nucleotide polymorphism (SNPs) in the 129/Sv and Cast/Ei lines was downloaded from the Sanger institute (v.5 SNP142)[52]. These were used as input for SNPsplit v0.3.4[53], to construct an N-masked reference genome based on mm10 in which all SNPs between 129/Sv and Cast/Ei were masked. The 50 bp single-end RNA-seq and 50 bp paired-end ChIP-seq reads were mapped to this N-masked reference genome using the default settings of hisat2 v2.2.1 and bowtie2 v2.4.1, respectively[54,55]. SNPsplit (--paired for the ChIP-seq analysis) was then used to assign the reads to either the 129/Sv or Cast/Ei bam file based on the best alignment or to a common

bam file if mapping to a region without allele-specific SNPs. The allele-specific and unassigned bam files were sorted using samtools v1.10[56].

For the RNA-seq, the number of mapped reads per gene were counted for both alleles separately using HTSeq v0.12.4 (--nonunique=none -m intersection-nonempty)[57] based on the gene annotation from ensembl v98. For each condition, genes with more than 20 allele-specific reads across both replicates were used to calculate the allelic ratio, defined as $\left(\frac{Xi}{Xi+Xa}\right)$. For day 0 and day 7, Cast/Ei and 129/Sv were used as the Xi and active allele (Xa), respectively, whereas for day 3, Cast/Ei and 129/Sv were used as Xa and Xi, respectively. The allelic ratios of X-linked genes were visualized as violin plots with boxplots of the same data on top. Significant differences between conditions were tested using a two-sided Mann–Whitney test with α < 0.05. P-values were corrected for multiple testing using the Benjamini–Hochberg procedure. Different conditions were compared by plotting the X-linked genes that had more than 20 allele-specific reads in both conditions along the X chromosome. A two-sided Wilcoxon signed-rank test with α < 0.05 was used to test for significant differences between the allelic ratios of the different conditions, after which P-values were corrected for multiple testing using the Benjamini–Hochberg procedure. We visualized differences in allelic ratios of X-linked genes between conditions by plotting both ratios on the different axes of a scatter plot. Genes were highlighted when they were identified as lowly silenced genes in *Spen⁻/⁻* ESCs from[27], which were defined as genes showing $-0.05 < z < -0.2$ where z (gene silencing) $= \left(\frac{Xi}{Xi+Xa}\right)dox - \left(\frac{Xi}{Xi+Xa}\right)noDox$.

The allele-specific ChIP-seq bam files were normalized using the 'callpeak' and 'bdgcmp' functions of MACS2 v2.2.7.1[58]. We called broad peaks (-f BAMPE --broad --bdg) and used the Poisson P-value as the method for normalizing the tracks. The input-normalized tracks were visualized using pyGenomeTracks v3.4[59]. For validation, we downloaded several publicly available datasets. The SPEN CUT&RUN data (GSE131782, samples: SRX5903674, SRX5903675, SRX5903676, SRX5903677, SRX5903678, SRX5903679, SRX5903682, and SRX5903683)[25], were processed similar to our analysis using a C57BL/6NJ-Cast/Ei reference genome. However, the allele-specific tracks were normalized based on the total number of mapped reads per sample. The scaling factor was calculated as 10^6 / total number of mapped reads and used as parameter --scaleFactor to both allelic tracks using deepTools bamCoverage v3.5.0.[60]. A binsize of 1 was used and paired-end reads were extended. The allele-specific tracks from HDAC3 and H3K27Ac (GSE116990, samples: SRX4384412, SRX4384420, SRX4384476, SRX4384484, SRX4887836, and SRX4887839) were downloaded from[29]. For all datasets, replicates for each condition were averaged using deepTools bigwigCompare v.3.5.0 with the settings '--operation mean --binSize 1'[60]. In the genome browser overview showing the allele-specific tracks, the y axis was scaled for each group of samples separately.

**Reporting summary**. Further information on research design is available in the Nature Research Reporting Summary linked to this article.

## Data availability
All raw and processed high-throughput sequencing data (ChIP-seq, RNA-seq) generated in this study have been submitted to the NCBI Gene Expression Omnibus (GEO) under accession number GSE163321. All other relevant data are available in the Supplementary Information, Source Data file, or from the corresponding author upon request. We used data from the following publicly available datasets: GSE131782 and GSE116990. Source data are provided with this paper.

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

## Acknowledgements

We thank Anniek Meesters and Esther Sleddens-Linkels for technical help; Kristian Helin for providing the EZH2 antibody; Martine M. Jaegle for sharing with us a floxed-puromycin resistance cassette; Joana Carvalho Moreira de Mello for her input in R plotting; and all the members of the Erasmus MC Developmental Biology department for useful discussions. T.R.F. is supported by an Erasmus MC grant (Mrace). T.R.F., B.F.T., H.M.B., E.T., and J.G. are supported by the Oncode Institute and a ZonMW Top subsidy (91215046). B.D.G. is supported by a research grant of the University Medical Center Giessen and Marburg (UKGM) and by a Prize of the Justus Liebig University Giessen. The work was further supported by the Deutsche Forschungsgemeinschaft (DFG, German Research Foundation)—TRR 81/3-109546710 and BO1639/9-393040308, and the Von Behring-Röntgen foundation and Excellence Cluster for Cardio Pulmonary System (ECCPS) in Giessen to T.B.

## Author contributions

T.R.F. and J.G. conceived the project and designed the experiments. T.R.F. and E.T. performed most of the experimental work and data analysis. H.M.B. performed the ChIP-seq experiments. B.F.T. performed all the bioinformatic analysis. C.G. helped setup the high-molecular-weight Western blot analysis. S.M. setup the allele-specific RT-qPCR analysis. B.D.G., F.D., T.B. and E.H. provided valuable resources and input. H.M.B. and J.G. supervised the work. T.R.F. and J.G. wrote the paper with input from all the authors. H.M.B. helped review and edit the paper. J.W.M.M. and J.G. were responsible for the funding acquisition. RNA-seq and ChIP-seq datasets were generated in the Erasmus MC Center for Biomics led by W.F.J.v.IJ.

## Competing interests

The authors declare no competing interests.
