## [Peer Review File · Nature Communications]

Title: SPEN is Required for Xist Upregulation during Initiation of X Chromosome InactivationREVIEWER COMMENTS

Reviewer #1 (Remarks to the Author):

Robert-Finestra et al., investigate in detail the function of SPEN in mouse XCI. By generating Spen knockout mouse ES cells clones, they uncover that lack of Spen prevents Xist upregulation upon differentiation leading to defective gene silencing. Functionally they suggest that Spen negatively regulates Tsix expression, and is therefore needed to release Tsix repression of Xist expression. This work is a logical extension of previous publications^{1–3} about the relationship between Spen and Xist, and further clarifies the role of Spen as a regulator of Tsix as well as acting to stabilize the Xist RNA during initiation of XCI.

Overall, the authors' work and experiments are relevant and the is article well-written. There are minor comments that would be needed to be address before publication.

Minor comments:

1. Many of the figures (both main and supplementary) have different formatting and it is suggested that the authors change it for consistency and to facilitate reading the article. Some of the colors chosen for the graphs may be hard to read for someone that has colorblindness and the authors may consider changing those.

a. Fig. 1h: Tsix pinpoints could be indicated by an arrowhead

b. Fig. 2b: For consistency, the heading reading Xist/DAPI could be flipped 180°

c. Fig. 3b: Colors in the graph is not explained

2. There are some figures where statistics are not applied to the data.

a. Supp. Fig. 3d: is it possible to show some statistics for the indicated partial rescue of the silencing effect that is seen at day 5?

3. In the methods section there is missing information about what companies the reagents come from and what catalog number they have. There are also missing information in how some experiments we done.

a. The cloning procedure of gRNAs into pX458/459 is not explained

b. Under the RNA stability assay it says that feeders are removed by placing the cells on non-gelatinized plates. Can you be sure that you do not have any remaining contaminating feeders in your sample? Would it be possible to distinguish feeder signals from true sample signals? This could also be explained better if information about the feeders were provided

c. On row 5 under RT-PCR it says "were" where it should be "where"

d. On row 4 under IF it says "antibodies" where it should be "antibody".

e. The RNA seq method chosen for this study was Smart-seq2, which originally is a protocol for single-cell sequencing. From what we have understood the authors have performed bulk RNAseq and can therefore not simply state that they followed the original article's protocol. We suggest that the authors add information of how they collected the cells and processed them to generate cDNA libraries

- f. On row 9 under ChIP seq it says “1µg antibody” but not the name of the antibody
- g. “ChIP-seq library was prepared following the manufacturer’s instructions”, would need a brief summary of the steps in the manufacturer’s protocols

4. Nature communications reaches a diverse audience, therefore the authors should discuss how their work can translate into human relevance since TSIX regulatory regions are missing in humans⁴, and also considering the amount of studies suggesting XCI in mouse is quite different from human.

Comments on language and article structure

Overall, this is a well-written article where the introduction follows a funnel structure. The results are generally clearly written but there are some sentences that would benefit from being placed in the discussion, which also should start with their conclusions/findings and not with another introduction.

5. There are parts in the text in the results section that are discussing the results rather than presenting them and the authors should consider moving these to the discussion instead for better flow.

a. “This abnormal cloud formation could be related to higher SPEN abundance due to overexpression in the undifferentiated state, possibly stabilizing Xist or silencing Tsix. In addition, these results indicate that the observed defect in Xist expression is SPEN mediated and takes place at the very early initiation steps of XCI”

b. “Our results show that SPEN accumulation on the Xi can be distinguished in an early state and late state of SPEN accumulation, where the early state is marked by small dispersed accumulation of SPEN co-localizing with EZH2 without detectable H3K27me3 at day 2, whereas the late state SPEN accumulation is compact and co-localizes with EZH2 and H3K27me3”

c. “This finding could be explained by a role for SPEN in stabilizing Xist RNA by complex formation”

d. “However, these results cannot explain why physiological Xist upregulation is lost in Spen^{-/-} ESCs upon differentiation”

e. “This Tsix regulatory region comprises the minor and major Tsix promoters and Xite, an enhancer of Tsix. Xist-mediated recruitment of SPEN is important for recruitment and/or activation of HDAC3, responsible for the removal of H3K27ac from the future Xi”

6. The beginning of the discussion should be changed because it starts with another introduction that also suggests that what they show in the article is still unknown.

a. The role of the silencing factor SPEN in the establishment phase of XCI has been addressed in various studies, but whether SPEN is relevant for the initiation phase, involving Xist upregulation, remains unknown.

References

1. McHugh, C. A. et al. The Xist lncRNA interacts directly with SHARP to silence transcription through HDAC3. *Nature* 521, 232–236 (2015).
2. Monfort, A. et al. Identification of Spen as a Crucial Factor for Xist Function through Forward Genetic Screening in Haploid Embryonic Stem Cells. *Cell Rep* 12, 554–61 (2015).

3. Chu, C. et al. Systematic discovery of Xist RNA binding proteins. *Cell* 161, 404–16 (2015).
4. Migeon, B. R., Lee, C. H., Chowdhury, A. K. & Carpenter, H. Species Differences in TSIX/Tsix Reveal the Roles of These Genes in X-Chromosome Inactivation. *Am J Hum Genet* 71, 286–293 (2002).

Reviewer #2 (Remarks to the Author):

In this paper, the authors investigate the role of SPEN in the process of X-chromosome inactivation using mouse embryonic stem cells from a hybrid genetic background, which allows for precise allelic analysis. They proceeded to generate SPEN^{+/-} and SPEN^{-/-}, in both wt mESC and in cells where the endogenous promoter is replaced by a dox-inducible promoter. Using these cellular models, they confirm that SPEN is, in a dose-dependent manner, essential for X-chromosome inactivation to proceed upon exit of pluripotency, but provide novel evidence that Xist expression and stability in itself depends on the presence of SPEN, both from the wt allele and from the dox-inducible one. Exogenous overexpression of Spen cDNA in these Spen^{-/-} cells rescues Xist expression from the wt allele. The authors then show that in absence of SPEN, Tsix expression is not properly silenced and propose that sustained Tsix expression is responsible for improper Xist expression in SPEN mutant. Finally, immunofluorescence analysis of SPEN localization using a GFP fusion protein reveals early and sustained recruitment of this protein, starting at the onset of Xist expression. Although the experiments presented are of high quality and I must acknowledge that they use numerous clever genetic constructs to address the biology of SPEN, I have several major concerns with the manuscript as it currently stands that need to be somehow address for this manuscript to convey its full potential and be suitable for publication. First and foremost, reading the manuscript gives the impression of a patchwork not successfully integrated, because the authors switch from Xist regulation, to SPEN accumulation back to Xist regulation looking at Tsix, and this introduces more confusion than necessary. I hope some of the following comments/suggestions will allow the author to address this issue.

My perception is that the main novel finding of this article is the early relationship between SPEN and Xist, which insures that Xist is timely expressed, a crucial parameter for the induction of XCI. This seems quite robust and reproducible in the authors set up as even dox-induced Xist is not produced, even though the inducible promoter yields more than 10x more Xist RNA than the endogenous promoter. Although I fully appreciate the importance of Tsix on the regulation of endogenous Xist expression, and convincingly show that interference with Tsix expression rescues Xist expression. The authors however further suggest that SPEN is important for the stability of Xist RNA without assessing this in these Tsix-stop cells, but rather in simple Spen KO cells. This is problematic as comparing the stability of RNA molecules which are not even remotely present in equivalent amounts. Importantly although sustained, but still significantly reduced, Tsix transcription may indeed interfere with endogenous Xist expression, the authors don't address the underpinnings of the impact of SPEN deletion on dox-induced Xist. This is a central issue that needs to be investigated here, especially since not all studies investigating the Xist-SPEN relationship have picked up this effect on Xist, but that the cited paper, which also identified this effect, used a dox-induced transgene on chromosome 3. All in all, the role of Tsix need to be further address: the author's claims in the discussion far exceed what is actually demonstrated in the data: "In

Spn^{-/-} cells, we detect higher levels of Tsix at the latest time points of monolayer differentiation, compared to Wt cells. In line with this, paternal Tsix levels in female E3.5 blastocysts with a mutated Xist RepA, necessary for SPEN recruitment, are higher than in Wt blastocysts³⁸, suggesting that higher Tsix levels in Spn^{-/-} cells are due to defective Tsix silencing, rather than a lack of Xist antisense transcription.” Figure 1J does not support at all this claim, Tsix is not higher in absence of SPEN, and is reduced during differentiation, albeit not to normal level. In any case, the effective molecular action of SPEN on Xist needs to be more precisely addressed before publication can be considered. Another concern I have is with the with the interpretation of “two distinguishable states” of SPEN accumulation. To this reviewer, SPEN accumulation follows Xist accumulation and other molecular events then occur. SPEN is not preloaded on the X, nor on Xist locus prior to its induction. The fact that EZH2 and H3K27me3 appears after Xist localization does not equate two distinguishable states of XIST accumulation. Importantly, even though I missed the point, I fail to understand how this is relevant in this manuscript.

Minor points

Figure 1b. The scheme is not useful and does not properly depict the extent of Dox treatment before d0.
Figure 1c. The level of detail given in these panels is not really useful for the purpose of the author and would probably be better conveyed with a violin plot showing the ratios for the different conditions used. To make the point that SPEN depletion affects XCI over the whole X, a simple panel in the supplementary figure would suffice.

Figure 1d: The color coding could be more marked for better readability.

Difference between figure 4a and 1e? I deduced the same cells were used in both setting but there is a 50x difference between the detected levels of Xist.

When relative RNA levels are presented, it would be appreciated to have a mention in the legend to which normalizer/condition.

Reviewer #3 (Remarks to the Author):

In this interesting manuscript the authors show that SPEN, a protein known to be recruited by Xist in the context of X inactivation, has a dual function required to both silence the antisense of Xist, Tsix, and to facilitate Xist upregulation and stabilization of its RNA. The finding of a dual function of SPEN is original. They identified these functions in a series of well thought out and well controlled experiments on embryonic stem cells in which Spn was deleted. They first used cells with an inducible Xist, but importantly also confirmed their results using female cells with a normal pattern of differentiation. One important and novel finding is the existence of early and late states of SPEN accumulation on the inactive X chromosome.

Using Spn^{-/-}/Tsix-stop female mouse embryonic stem cells, the authors also found that SPEN-mediated silencing of the Tsix promoter is required for Xist upregulation and this process happens very early in

XCI. The authors performed SPEN rescue experiments to confirm their observations. They further explored the occupancy profiles of multiple epigenetic markers including H3K27me3, H3K27ac and HDAC to investigate Tsix regulatory mechanism. Overall, the authors provide solid experimental data to support their claims and statistical analysis is appropriate. Clarification of the model presented should be provided.

Major comments,

1. As stated above an important observation the authors made is the existence of two distinguishable SPEN accumulation states. The authors should provide more quantitative measurements to distinguish early and late accumulations. For example, the authors could measure the intensity of the SPEN accumulation cluster. This data is critical for the following examination of co-localization of EZH2 and H3K27me3, which should also be measured.

2. The authors claimed that SPEN dosage is important in XCI as they observed reduced Rnf12 silencing in *Spen*^{+/-} cells. SPEN is a transcriptional repressor and is involved in various biological processes during differentiation. The authors should examine the possibility that this partial silencing may be due to a disruption in differentiation. Why are the levels of Nanog and Gata5 higher in *Spen* mutants? Could the authors comment on this finding?

3. Figure 6 and its legend are not clear. The figure should be better described in the Discussion. What are the states represented by the boxes? Why are there two examples of X-linked genes? Is the one on the right a gene that escape XCI? This is not mentioned in the text. Why is that gene associated with Xist stability? If the grey circles are RNA polymerase do they mark the escape gene? I presume that wiggly lines over the genes represent the transcripts?

Minor comments,

4. The authors show that *Spen*^{-/-} cells have a lower Xist expression than WT cells during the differentiation process regardless of Dox treatment in Figure 1e and 1g. In contrast *Spen*^{+/-} (heterozygous) cells present a higher level of Xist expression than WT cells at day 7 with or without Dox treatment in Supplementary Figure 2. The authors should provide a potential explanation.

5. The authors show that they can rescue Xist expression levels by introduction of full-length SPEN cDNA. They confirm the *Spen* RNA level by PCR. Could the authors examine the protein level to confirm that the SPEN protein level is also rescued?

6. The authors noticed the rescued cells have a higher Xist expression and also abnormal Xist cloud coating at day 0. It is possible that the overexpression of SPEN leads to defective Tsix silencing, as the authors stated. It is also possible that the ES cells undergo differentiation during screening of the rescued cells. The authors should examine the expression of pluripotency marker genes in the rescued cells.

7. Figure 3b legend should indicate what light and dark green color represent.

We thank the reviewers for their insightful comments and suggestions to improve our manuscript. We believe the changes we did make this a more solid and comprehensible manuscript. As described below we have been able to address most of the points risen by the reviewers. Below we answer all the comments point-by-point and have adapted the manuscript accordingly, all the changes in the manuscript file are highlighted in red.

Reviewer #1 (Remarks to the Author):

Robert-Finestra et al., investigate in detail the function of SPEN in mouse XCI. By generating Spen knockout mouse ES cells clones, they uncover that lack of Spen prevents Xist upregulation upon differentiation leading to defective gene silencing. Functionally they suggest that Spen negatively regulates Tsix expression, and is therefore needed to release Tsix repression of Xist expression. This work is a logical extension of previous publications 1–3 about the relationship between Spen and Xist, and further clarifies the role of Spen as a regulator of Tsix as well as acting to stabilize the Xist RNA during initiation of XCI.

Overall, the authors' work and experiments are relevant and the is article well-written. There are minor comments that would be needed to be address before publication.

Minor comments:

1. Many of the figures (both main and supplementary) have different formatting and it is suggested that the authors change it for consistency and to facilitate reading the article. Some of the colors chosen for the graphs may be hard to read for someone that has colorblindness and the authors may consider changing those.

We thank this reviewer for the suggestions. We tried to homogenize the format of all figures. We have also modified those figures that contained red and green or other difficult colors to distinguish and changed the original colors to a colorblind-friendly palette.

a. Fig. 1h: Tsix pinpoints could be indicated by an arrowhead

We have indicated *Xist* clouds with a white asterisk and *Tsix* pinpoints with a white arrowhead in one cell per image. Please note that Fig. 1h is now **Fig. 1g**.

b. Fig. 2b: For consistency, the heading reading Xist/DAPI could be flipped 180°

Adapted accordingly. Also in **Fig. 6d**.

c. Fig. 3b: Colors in the graph is not explained

Since we have adapted this figure, the figure legend is no longer required.

2. There are some figures where statistics are not applied to the data.

a. Supp. Fig. 3d: is it possible to show some statistics for the indicated partial rescue of the silencing effect that is seen at day 5?

We have performed a Mann-Whitney test for unpaired samples to compare *Spn*^{-/-} and *Spn* rescue (*Spn*^{-/-}(cDNA)) ESCs, they are significantly different at day 0 and 5. Note that this figure is now **Supplementary Fig. 3e**.

3. In the methods section there is missing information about what companies the reagents come from and what catalog number they have. There are also missing information in how some experiments we done.

We have added the appropriate information and for those reagents missing the company name and catalog number have been included in the revised manuscript.

a. The cloning procedure of gRNAs into pX458/459 is not explained.

We have explained the sgRNAs cloning protocol in the “Gene editing using the CRISPR/Cas9 technology” section (Methods).

b. Under the RNA stability assay it says that feeders are removed by placing the cells on non-gelatinized plates. Can you be sure that you do not have any remaining contaminating feeders in your sample? Would it be possible to distinguish feeder signals from true sample signals? This could also be explained better if information about the feeders were provided

Growing female mouse ESCs on male Mouse Embryonic Fibroblast (MEFs) to study XCI, taking advantage that male cells cannot undergo XCI, is common practice in the XCI field¹⁻³. The most cost effective way to eliminate the feeders is to plate the ESC + MEF culture on non-gelatinized plates, where only the MEFs attach. We are certain that only a small percentage of MEFs remains in the ESCs fraction, which will have a minimal impact in any downstream assay, like RT-qPCR or FISH. A possible alternative would have been to grow the ESCs in feeder-free conditions using 2i + LIF, but previous studies performed in our laboratory show that these conditions change the transcriptional state of the X Inactivation Center, that directly affects XCI initiation⁴. Therefore, we consider that using male MEFs and eliminating them via plating the cells in non-gelatinized plates is the best option for our study. In the first sentence of the Cell culture section (Methods) we have included detailed information about the feeder cells used in this study.

c. On row 5 under RT-PCR it says “were” where it should be “where”

Adapted accordingly.

d. On row 4 under IF it says “antibodies” where it should be “antibody”.

Adapted accordingly.

e. The RNA seq method chosen for this study was Smart-seq2, which originally is a protocol for single-cell sequencing. From what we have understood the authors have performed bulk RNAseq and can therefore not simply state that they followed the original article’s protocol. We suggest that the authors add information of how they collected the cells and processed them to generate cDNA libraries

The smart-seq2 method used for RNA sequencing in our study was, indeed, originally published as a single-cell protocol. In our study, we forced *Xist* upregulation from the endogenous *Xist* locus using a bidirectional doxycycline promoter that produces dsRed (**Fig. 1a**)⁵. Thus, for those conditions in which we forced *Xist* upregulation (**Supplementary Fig. 1d**), we isolated dsRed-positive cells by FACS sorting. In some of the conditions we obtained a low number of cells, therefore, we used an adapted version of the smart-seq2 able to create high-quality DNA libraries from about 1000 cells. A maximum of 1 ng of total RNA was used in the original smart-seq2 procedure without modifications. Although this method is published for single-cell experiments, it is capable of generating high quality cDNA from more than single cells. Bioanalyzer plots of the cDNA generated in this study show the library profile indicated in the Fig. 2a of the smart-seq2 paper for good libraries⁶. We have used before the smart-seq2 procedure for cell numbers below 1000 cells and obtained high quality transcriptome data as previously published⁷⁻⁹. In the revised manuscript, we have added a clarification regarding this question in the RNA-seq section (Methods).

f. On row 9 under CHIP seq it says “1µg antibody” but not the name of the antibody

We apologize and now refer to the H3K27me3 antibody in the text.

g. “ChIP-seq library was prepared following the manufacturer’s instructions”, would need a brief summary of the steps in the manufacturer’s protocols

We have added additional and more detailed information on the Chip-seq library preparation steps in the appropriate section.

4. Nature communications reaches a diverse audience, therefore the authors should discuss how their work can translate into human relevance since TSIX regulatory regions are missing in humans⁴, and also considering the amount of studies suggesting XCI in mouse is quite different from human.

We thank this reviewer for the suggestion and in our revised manuscript we cover the translational aspects of our findings in the last sentences of the discussion.

Comments on language and article structure

Overall, this is a well-written article where the introduction follows a funnel structure. The results are generally clearly written but there are some sentences that would benefit from being placed in the discussion, which also should start with their conclusions/findings and not with another introduction.

We have tried to adapt our manuscript according to the reviewers suggestion (see below).

5. There are parts in the text in the results section that are discussing the results rather than presenting them and the authors should consider moving these to the discussion instead for better flow.

a. "This abnormal cloud formation could be related to higher SPEN abundance due to overexpression in the undifferentiated state, possibly stabilizing Xist or silencing Tsix. In addition, these results indicate that the observed defect in Xist expression is SPEN mediated and takes place at the very early initiation steps of XCI"

b. "Our results show that SPEN accumulation on the Xi can be distinguished in an early state and late state of SPEN accumulation, where the early state is marked by small dispersed accumulation of SPEN co-localizing with EZH2 without detectable H3K27me3 at day 2, whereas the late state SPEN accumulation is compact and co-localizes with EZH2 and H3K27me3

c. "This finding could be explained by a role for SPEN in stabilizing Xist RNA by complex formation"

d. "However, these results cannot explain why physiological Xist upregulation is lost in Spen^{-/-} ESCs upon differentiation"

e. "This Tsix regulatory region comprises the minor and major Tsix promoters and Xite, an enhancer of Tsix. Xist-mediated recruitment of SPEN is important for recruitment and/or activation of HDAC3, responsible for the removal of H3K27ac from the future Xi"

We understand the concern of the reviewer, as some of these sentences are a bit speculative and could go to the discussion, therefore we have reformulated statement a, b and c. However, statements d and e introduce the following results section and, therefore, decided to leave these in place.

6. The beginning of the discussion should be changed because it starts with another introduction that also suggests that what they show in the article is still unknown.

a. The role of the silencing factor SPEN in the establishment phase of XCI has been addressed in various studies, but whether SPEN is relevant for the initiation phase, involving Xist upregulation, remains unknown.

To avoid repetition, we have removed this sentence from the beginning of the discussion.

References

1. McHugh, C. A. et al. The Xist lncRNA interacts directly with SHARP to silence transcription through HDAC3. *Nature* 521, 232–236 (2015).
2. Monfort, A. et al. Identification of Spen as a Crucial Factor for Xist Function through Forward Genetic Screening in Haploid Embryonic Stem Cells. *Cell Rep* 12, 554–61 (2015).
3. Chu, C. et al. Systematic discovery of Xist RNA binding proteins. *Cell* 161, 404–16 (2015).
4. Migeon, B. R., Lee, C. H., Chowdhury, A. K. & Carpenter, H. Species Differences in TSIX/Tsix Reveal the Roles of These Genes in X-Chromosome Inactivation. *Am J Hum Genet* 71, 286–293 (2002).

Reviewer #2 (Remarks to the Author):

In this paper, the authors investigate the role of SPEN in the process of X-chromosome inactivation using mouse embryonic stem cells from a hybrid genetic background, which allows for precise allelic analysis. They proceeded to generate SPEN^{+/-} and SPEN^{-/-}, in both wt mESC and in cells where the endogenous promoter is replaced by a dox-inducible promoter. Using these cellular models, they confirm that SPEN is, in a dose-dependent manner, essential for X-chromosome inactivation to proceed upon exit of pluripotency, but provide novel evidence that Xist expression and stability in itself depends on the presence of SPEN, both from the wt allele and from the dox-inducible one. Exogenous overexpression of Spen cDNA in these Spen^{-/-} cells rescues Xist expression from the wt allele. The authors then show that in absence of SPEN, Tsix expression is not properly silenced and propose that sustained Tsix expression is responsible for improper Xist expression in SPEN mutant. Finally, immunofluorescence analysis of SPEN localization using a GFP fusion protein reveals early and sustained recruitment of this protein, starting at the onset of Xist expression. Although the experiments presented are of high quality and I must acknowledge that they use numerous clever genetic constructs to address the biology of SPEN, I have several major concerns with the manuscript

as it currently stands that need to be somehow address for this manuscript to convey its full potential and be suitable for publication. First and foremost, reading the manuscript gives the impression of a patchwork not successfully integrated, because the authors switch from Xist regulation, to SPEN accumulation back to Xist regulation looking at Tsix, and this introduces more confusion than necessary. I hope some of the following comments/suggestions will allow the author to address this issue.

We thank this reviewer for the positive and constructive comments. With respect to the buildup and flow of the manuscript we believe our manuscript follows a logical order. We start presenting our main finding that “*Spn*^{-/-} ESCs are unable to upregulate *Xist* upon differentiation”, which suggests that SPEN plays a role early in XCI. We further investigate this observation by interrogating SPEN-GFP accumulation to the inactive X (Xi) chromosome upon differentiation. After confirming that SPEN is present early during differentiation on the Xi, we set out to explore the mechanism behind the lack of *Xist* upregulation, by studying *Xist* RNA stability and the regulatory role of SPEN in *Tsix* repression. To us this is a coherent flow of ideas to narrate our story. We think we further improved the flow of the story by dividing Fig. 5 into two different figures (**Fig. 5** and **Fig. 6**) and other minor adjustments.

My perception is that the main novel finding of this article is the early relationship between SPEN and Xist, which insures that Xist is timely expressed, a crucial parameter for the induction of XCI. This seems quite robust and reproducible in the authors set up as even dox-induced Xist is not produced, even though the inducible promoter yields more than 10x more Xist RNA than the endogenous promoter.

We would like to clarify that *Spn*^{-/-} doxycycline-inducible *Xist* ESCs are able to produce *Xist* RNA (**Fig. 1d**), although at lower levels than in the Wt situation, same as shown in Nesterova et al., 2019. While, *Spn*^{-/-} ESCs upon monolayer differentiation without forcing *Xist* upregulation (no doxycycline) cannot upregulate *Xist* (**Fig. 1f**).

Although I fully appreciate the importance of Tsix on the regulation of endogenous Xist expression, and convincingly show that interference with Tsix expression rescues Xist expression. The authors however further suggest that SPEN is important for the stability of Xist RNA without assessing this in these Tsix-stop cells, but rather in simple *Spn* KO cells. This is problematic as comparing the stability of RNA molecules which are not even remotely present in equivalent amounts.

We thank the reviewer for this suggestion and addressed this issue in **Fig. 6f**, where we have interrogated *Xist* RNA stability in differentiating (day 3) Wt and *Spn*^{-/-} *Tsix*.Stop ESCs using actinomycin D. In this setup in the Wt and knockout situation we observe comparable *Xist* RNA levels (**Fig. 6c**). In

Tsix.Stop ESCs, we observe that *Xist* RNA in *Spem*^{-/-} ESCs is less stable than in the Wt situation, same as shown in **Fig. 4d**, were we determined the *Xist* RNA stability in doxycycline-inducible ESCs upon 4 days of doxycycline treatment. Moreover, a very recent publication described, using a different approach, namely microscopy, that SPEN plays a role in *Xist* RNA stability¹¹, reinforcing our results as we describe in the discussion.

Importantly although sustained, but still significantly reduced, *Tsix* transcription may indeed interfere with endogenous *Xist* expression, the authors don't address the underpinnings of the impact of SPEN deletion on dox-induced *Xist*. This is a central issue that needs to be investigated here, especially since not all studies investigating the *Xist*-SPEN relationship have picked up this effect on *Xist*, but that the cited paper, which also identified this effect, used a dox-induced transgene on chromosome 3.

Nesterova et al., 2019 show that *Xist* RNA levels in *Spem*^{-/-} ESCs are reduced when the authors force *Xist* upregulation from a doxycycline-inducible transgene in chromosome 3 (Ch3) and also using a doxycycline-inducible promoter replacing the endogenous *Xist* promoter in the X chromosome (ChX). In the first case *Tsix* does not play a role, while in the second case *Tsix* is functional. Nesterova et al., 2019 Supplementary Fig. 2d shows by *Xist* RNA FISH analysis in the Ch3 situation that the percentage of *Xist* clouds per nuclei is only slightly reduced compared to the Wt situation, while at the endogenous *Xist*-inducible transgene the percentage of *Xist* clouds per nuclei is considerably lower than the Wt situation and the cloud morphology is different, i.e. the authors describe the clouds as "weak and diffuse". In our study, we also observe lower number of *Xist* clouds, that are in general smaller, in *Spem*^{-/-} compared to Wt ESCs upon forced *Xist* upregulation from the endogenous locus (**Fig. 4b,c**). To us this suggests that in the Ch3 transgene, the observed effect is caused by the role of SPEN in *Xist* RNA stability. While in the setting of the doxycycline-inducible promoter at the endogenous locus, the phenotype is more severe as not only *Xist* stability is affected, but also *Tsix* is not properly silenced, affecting *Xist* cloud formation and morphology more severely.

We believe that the most relevant setting to study the relationship between SPEN and *Tsix* is in a physiological context as is studied herein. Therefore, in our manuscript we studied the SPEN and *Tsix* relationship using hybrid F1:129/Cast, *Tsix*.Stop and *Tsix*.Cherry ESC lines (**Fig. 6** and **Supplementary Fig. 6**), rather than studying cell lines with a doxycycline-inducible system.

All in all, the role of *Tsix* need to be further address: the author's claims in the discussion far exceed what is actually demonstrated in the data: "In *Spem*^{-/-} cells, we detect higher levels of *Tsix* at the latest time points of monolayer differentiation, compared to Wt cells. In line with this, paternal *Tsix* levels in female E3.5 blastocysts with a mutated *Xist* RepA, necessary for SPEN recruitment, are higher than in

Wt blastocysts³⁸, suggesting that higher *Tsix* levels in *Spn*^{-/-} cells are due to defective *Tsix* silencing, rather than a lack of *Xist* antisense transcription.” Figure 1J does not support at all this claim, *Tsix* is not higher in absence of SPEN, and is reduced during differentiation, albeit not to normal level. In any case, the effective molecular action of SPEN on *Xist* needs to be more precisely addressed before publication can be considered.

Please note that Fig. 1j is now Fig. 1i. To interpret Fig. 1i we should take into consideration that *Tsix* is regulated by different pluripotency factors like OCT4, SOX2, KLF4, c-MYC and REX1¹²⁻¹⁴. In this work we identify SPEN as new *Tsix* regulator. In Fig. 1i we can see how upon monolayer differentiation (from day 0 to 3) Wt and *Spn*^{-/-} ESCs are able to downregulate *Tsix* from the 129 (chromosome undergoing XCI) and Cast allele. This reduction is most likely caused by the aforementioned *Tsix* regulators, that are linked to pluripotency. From day 3 onwards, the two alleles act differently: in the Cast allele *Tsix* levels remain constant in the Wt and knockout situation, while in the 129 allele, Wt ESCs are able to further silence *Tsix* expression, while *Spn*^{-/-} cells cannot, showing that these differences in *Tsix* levels are caused by SPEN. Therefore, we can affirm that the 129 *Tsix* levels are higher in *Spn*^{-/-} ESCs upon differentiation (day 3, 5 and 7) compared to Wt cells. Similar results are shown in **Supplementary Fig. 3f** and **Supplementary Fig. 6c**. Besides, the reason we do not find a difference at day 0 is related to the fact that *Xist* is not yet upregulated and therefore *Xist* mediated recruitment of SPEN is absent.

Moreover, we have modified the cited sentences to make them less speculative (see discussion section).

Another concern I have is with the interpretation of “two distinguishable states” of SPEN accumulation. To this reviewer, SPEN accumulation follows *Xist* accumulation and other molecular events then occur. SPEN is not preloaded on the X, nor on *Xist* locus prior to its induction. The fact that EZH2 and H3K27me3 appears after *Xist* localization does not equate two distinguishable states of XIST accumulation. Importantly, even though I missed the point, I fail to understand how this is relevant in this manuscript.

We thank the reviewer for this comment. In the initial submission of our manuscript, we distinguished between early and late SPEN accumulation, based on the immunofluorescence intensity and compaction state. We have tried to quantify this distinction, by measuring the intensity of the SPEN-GFP accumulations over time (Fig. 3c), although we didn’t quantify the compaction state, since we couldn’t find a reliable method to measure it. In any case, these results show an increase in SPEN-GFP signal intensities over time, but we do not see a distribution that supports our initial distinction between early and late SPEN accumulation. Based on these observations, we have decided to remove

the difference between early and late SPEN accumulation and merge them into one group of total SPEN-GFP accumulation. Nevertheless, this does not change the conclusion that SPEN is already recruited to the X chromosome at very early stages of XCI. We are sorry for the confusion and have adapted the text and figures accordingly, see **Fig. 3a-d,f** and **Supplementary Fig. 4e-g**.

Minor points

Figure 1b. The scheme is not useful and does not properly depict the extent of Dox treatment before d0.

We agree that the scheme was not completely clear and have removed it. We believe the text and figure legends are sufficient for the reader to understand the experimental design.

Figure 1c. The level of detail given in these panels is not really useful for the purpose of the author and would probably be better conveyed with a violin plot showing the ratios for the different conditions used. To make the point that SPEN depletion affects XCI over the whole X, a simple panel in the supplementary figure would suffice.

We thank this reviewer for the suggestion. To gain clarity, we have moved the violin plots to **Fig. 1** and moved the allelic ratio plots along the X chromosome to **Supplementary Fig. 1**.

Figure 1d: The color coding could be more marked for better readability.

Please note that Fig. 1d is now **Fig. 1c**. We have increased the color contrast between the 129 and Cast alleles in the aforementioned figure, as with, **Fig. 1e** and **Supplementary Fig. 2a,c, 3e** and **6d**.

Difference between figure 4a and 1e? I deduced the same cells were used in both setting but there is a 50x difference between the detected levels of *Xist*.

Please note that Fig. 1e is now **Fig. 1d**. Indeed both experiments were done with the same cells, but the primer pairs used in **Fig. 4a** and **1d** are not the same and therefore expression levels cannot be compared between figures. In **Fig. 1d**, we used allele-specific *Xist* primers to distinguish between 129 and Cast expression and used *Hist2h2aa1* gene as a reference gene. In **Fig. 4a**, where allele-specific information is not required, as we are interrogating total *Xist* RNA stability, we used non-allele specific primers for *Xist* RNA and *Beta-actin* as a reference gene. We show this difference by indicating “Relative total *Xist* expression” in the Y-axes. The reason why we use different reference genes is that to perform RNA stability assays with actinomycin D a very stable transcript is needed as a reference gene, that is the case of *Beta-actin*.

When relative RNA levels are presented, it would be appreciated to have a mention in the legend to which normalizer/condition.

We have adapted the figure legends and explained the reference gene used for each qPCR experiment. The same information is also explained in the methods section (RT-qPCR section).

Reviewer #3 (Remarks to the Author):

In this interesting manuscript the authors show that SPEN, a protein known to be recruited by Xist in the context of X inactivation, has a dual function required to both silence the antisense of Xist, Tsix, and to facilitate Xist upregulation and stabilization of its RNA. The finding of a dual function of SPEN is original. They identified these functions in a series of well thought out and well controlled experiments on embryonic stem cells in which Spen was deleted. They first used cells with an inducible Xist, but importantly also confirmed their results using female cells with a normal pattern of differentiation. One important and novel finding is the existence of early and late states of SPEN accumulation on the inactive X chromosome.

Using Spen^{-/-}/Tsix-stop female mouse embryonic stem cells, the authors also found that SPEN-mediated silencing of the Tsix promoter is required for Xist upregulation and this process happens very early in XCI. The authors performed SPEN rescue experiments to confirm their observations. They further explored the occupancy profiles of multiple epigenetic markers including H3K27me3, H3K27ac and HDAC to investigate Tsix regulatory mechanism. Overall, the authors provide solid experimental data to support their claims and statistical analysis is appropriate. Clarification of the model presented should be provided.

Major comments,

1. As stated above an important observation the authors made is the existence of two distinguishable SPEN accumulation states. The authors should provide more quantitative measurements to distinguish early and late accumulations. For example, the authors could measure the intensity of the SPEN accumulation cluster. This data is critical for the following examination of co-localization of EZH2 and H3K27me3, which should also be measured.

We thank the reviewer for this comment. In the initial submission of our manuscript, we distinguished between early and late SPEN accumulation, based on the immunofluorescence intensity and

compaction state. We have tried to quantify this distinction, by measuring the intensity of the SPEN-GFP accumulations over time (**Fig. 3c**), although we did not quantify the compaction state, since we couldn't find a reliable method to measure it. In any case, these results show an increase in SPEN-GFP signal intensities over time, but we do not see a distribution that supports our initial distinction between early and late SPEN accumulation. Based on these observations, we have decided to remove the difference between early and late SPEN accumulation and merge them into one group of total SPEN-GFP accumulation. Nevertheless, this does not change the conclusion that SPEN is already recruited to the X chromosome at very early stages of XCI. We are sorry for the confusion and have adapted the text and figures accordingly, see **Fig. 3a-d,f** and **Supplementary Fig. 4e-g**.

2. The authors claimed that SPEN dosage is important in XCI as they observed reduced *Rnf12* silencing in *Spn*^{+/-} cells. SPEN is a transcriptional repressor and is involved in various biological processes during differentiation. The authors should examine the possibility that this partial silencing may be due to a disruption in differentiation. Why are the levels of *Nanog* and *Gata5* higher in *Spn* mutants? Could the authors comment on this finding?

XCI and the pluripotency network are tightly linked. An example is that female ESCs with two active X chromosomes display a delay in differentiation¹⁵. In *Spn*^{-/-} ESCs, X-linked gene silencing is completely impaired, which could affect differentiation. Moreover, SPEN is a transcriptional repressor with diverse biological functions¹⁶⁻¹⁸, therefore, the absence of SPEN could potentially have indirect effects in the pluripotency state or differentiation potential. Thus, either one or a combination of the two explanations could be the cause of the aforementioned differences in pluripotency/differentiation markers between Wt and knockout ESCs (**Supplementary Fig. 1h**). Nonetheless, a strong evidence that *Spn*^{-/-} ESCs are able to upregulate *Xist*, initiate XCI and differentiation is that *Spn*^{-/-} *Tsix*.Stop ESCs can upregulate *Xist* to similar levels to Wt *Tsix*.Stop ESCs (**Fig. 6c**), despite not being able to silence the X-linked gene *Rnf12* (**Supplementary Fig. 6d**).

3. Figure 6 and its legend are not clear. The figure should be better described in the Discussion. What are the states represented by the boxes? Why are there two examples of X-linked genes? Is the one on the right a gene that escape XCI? This is not mentioned in the text. Why is that gene associated with *Xist* stability? If the grey circles are RNA polymerase do they mark the escape gene? I presume that wiggly lines over the genes represent the transcripts?

Please note that Fig. 6 is now **Fig. 7**. We have better integrated the figure and the legend by assigning each box a letter that is described in the figure legend. The two X-linked genes in the figure exemplify

that SPEN spreads together with *Xist* along the X chromosome silencing active genes, we did not mean to represent an escapee. We also wanted to show that SPEN is involved in *Xist* stability, independently of its function in repressing *Tsix*. We have modified the scheme for a more clear representation. Lastly, we have better integrated the graphical model in the discussion.

Minor comments,

4. The authors show that *Spn*^{-/-} cells have a lower *Xist* expression than WT cells during the differentiation process regardless of Dox treatment in Figure 1e and 1g. In contrast *Spn*^{+/-} (heterozygous) cells present a higher level of *Xist* expression than WT cells at day 7 with or without Dox treatment in Supplementary Figure 2. The authors should provide a potential explanation.

We are not certain why differentiating *Spn*^{+/-} ESCs display higher *Xist* expression levels compared to Wt ESCs, while X-linked gene silencing is reduced (**Supplementary Fig. 2 a-d**). A possible explanation could be that in differentiating *Spn* heterozygous ESCs *Xist* can be upregulated but cannot efficiently silence X-linked genes, as SPEN protein levels are dose dependent. This could create a feedback-loop that upregulates *Xist* even further.

5. The authors show that they can rescue *Xist* expression levels by introduction of full-length SPEN cDNA. They confirm the *Spn* RNA level by PCR. Could the authors examine the protein level to confirm that the SPEN protein level is also rescued?

We have addressed this point in **Supplementary Fig. 3d**, where we show a SPEN and FLAG Western blot of Wt, *Spn*^{-/-} and three rescue clones expressing the *Spn* cDNA from the ROSA26 locus. The Western blot clearly shows that the rescue clones express the SPEN protein. Moreover, only the rescue clones show the presence of the SPEN-FLAG tag present in the *Spn* cDNA expression construct. It is important to note that the *Spn* RT-qPCR on the same cells detects 2 to 3 times more *Spn* mRNA expression in the rescue clones compared to the Wt situation (**Supplementary Fig. 3c**). Moreover, we believe a Western blot for such a big protein (the predicted molecular weight of SPEN is approximately 400 kDa) is not sensitive enough to detect changes of this range, therefore we cannot draw conclusions on how SPEN protein levels exactly compare between conditions.

6. The authors noticed the rescued cells have a higher *Xist* expression and also abnormal *Xist* cloud coating at day 0. It is possible that the overexpression of SPEN leads to defective *Tsix* silencing, as the authors stated. It is also possible that the ES cells undergo differentiation during screening of the

rescued cells. The authors should examine the expression of pluripotency marker genes in the rescued cells.

We have addressed this point in **Supplementary Fig. 3g**, where we determined the expression levels of pluripotency markers *Sox2* and *Oct4* and the differentiation marker *Gata6* in Wt, *Spn*^{-/-} and the *Spn* rescue clones at day 0 and 5 of monolayer differentiation. We see that the levels of these markers at day 0 are comparable to those in the Wt and *Spn*^{-/-} situation, indicating that *Spn* overexpression does not force cells to differentiate.

7. Figure 3b legend should indicate what light and dark green color represent.

With the changes performed in **Fig. 3**, this legend is no longer necessary.

References

1. Gontan, C. *et al.* REX1 is the critical target of RNF12 in imprinted X chromosome inactivation in mice. *Nat. Commun.* **9**, 1–12 (2018).
2. Royce-Tolland, M. E. *et al.* The A-repeat links ASF/SF2-dependent Xist RNA processing with random choice during X inactivation. *Nat. Struct. Mol. Biol.* **17**, 948–54 (2010).
3. Colognori, D., Sunwoo, H., Wang, D., Wang, C.-Y. & Lee, J. T. Xist Repeats A and B Account for Two Distinct Phases of X Inactivation Establishment. *Dev. Cell* 1–12 (2020) doi:10.1016/j.devcel.2020.05.021.
4. Loos, F. *et al.* Xist and Tsix Transcription Dynamics Is Regulated by the X-to-Autosome Ratio and Semistable Transcriptional States. *Mol. Cell. Biol.* **36**, 2656–2667 (2016).
5. Loda, A. *et al.* Genetic and epigenetic features direct differential efficiency of Xist-mediated silencing at X-chromosomal and autosomal locations. *Nat. Commun.* **8**, (2017).
6. Picelli, S. *et al.* Smart-seq2 for sensitive full-length transcriptome profiling in single cells. *Nat. Methods* **10**, 1096–1100 (2013).
7. Crisan, M. *et al.* BMP signalling differentially regulates distinct haematopoietic stem cell types. *Nat. Commun.* **6**, (2015).
8. Crisan, M. *et al.* BMP and Hedgehog Regulate Distinct AGM Hematopoietic Stem Cells Ex Vivo. *Stem Cell Reports* **6**, 383–395 (2016).

9. Rao-Ruiz, P. *et al.* Engram-specific transcriptome profiling of contextual memory consolidation. *Nat. Commun.* **10**, 1–14 (2019).
10. Nesterova, T. B. *et al.* Systematic allelic analysis defines the interplay of key pathways in X chromosome inactivation. *Nat. Commun.* **10**, 1–15 (2019).
11. Rodermund, L. *et al.* Time-resolved structured illumination microscopy reveals key principles of Xist RNA spreading. *Science (80-.)*. **372**, eabe7500 (2021).
12. Donohoe, M. E., Silva, S. S., Pinter, S. F., Xu, N. & Lee, J. T. The pluripotency factor Oct4 interacts with Ctf and also controls X-chromosome pairing and counting. *Nature* **460**, 128–132 (2009).
13. Navarro, P. *et al.* Molecular coupling of Tsix regulation and pluripotency. *Nature* **468**, 457–460 (2010).
14. Gontan, C. *et al.* RNF12 initiates X-chromosome inactivation by targeting REX1 for degradation. *Nature* **485**, 386–390 (2012).
15. Schulz, E. G. *et al.* The two active X chromosomes in female ESCs block exit from the pluripotent state by modulating the ESC signaling network. *Cell Stem Cell* **14**, 203–216 (2014).
16. Oswald, F. *et al.* SHARP is a novel component of the Notch/RBP-Jk signalling pathway. *EMBO J.* **21**, 5417–5426 (2002).
17. Shi, Y. *et al.* Sharp, an inducible cofactor that integrates nuclear receptor repression and activation. *Genes Dev.* **15**, 1140–1151 (2001).
18. Kuroda, K. *et al.* Regulation of marginal zone B cell development by MINT, a suppressor of Notch/RBP-J signaling pathway. *Immunity* **18**, 301–312 (2003).

REVIEWERS' COMMENTS

Reviewer #1 (Remarks to the Author):

The authors have addressed all my comments in a satisfactory manner. Congratulations to a very nice paper!

Reviewer #2 (Remarks to the Author):

The new version of the manuscript has been vastly improved by numerous changes the author have made and I do not see the need for further changes.

Reviewer #3 (Remarks to the Author):

The authors clarified most of their conclusions and the manuscript is significantly improved. They also performed additional experiments to support their claims, including examination of the expression of Spen and analysis of some pluripotency marker genes. It is a bit disappointing that the authors could not quantify SPEN accumulation on the X, as the idea of two states was interesting. Nonetheless, the main novel conclusion that SPEN is recruited to the X at initiation of XCI is still valid.